# Oligodendrocytes and myelin limit neuronal plasticity in visual cortex

Wendy Xin[1✉], Megumi Kaneko[2], Richard H. Roth[3], Albert Zhang[1], Sonia Nocera[1], Jun B. Ding[3], Michael P. Stryker[2] & Jonah R. Chan[1✉]

Developmental myelination is a protracted process in the mammalian brain[1]. One theory for why oligodendrocytes mature so slowly posits that myelination may stabilize neuronal circuits and temper neuronal plasticity as animals age[2–4]. We tested this theory in the visual cortex, which has a well-defined critical period for experience-dependent neuronal plasticity[5]. During adolescence, visual experience modulated the rate of oligodendrocyte maturation in visual cortex. To determine whether oligodendrocyte maturation in turn regulates neuronal plasticity, we genetically blocked oligodendrocyte differentiation and myelination in adolescent mice. In adult mice lacking adolescent oligodendrogenesis, a brief period of monocular deprivation led to a significant decrease in visual cortex responses to the deprived eye, reminiscent of the plasticity normally restricted to adolescence. This enhanced functional plasticity was accompanied by a greater turnover of dendritic spines and coordinated reductions in spine size following deprivation. Furthermore, inhibitory synaptic transmission, which gates experience-dependent plasticity at the circuit level, was diminished in the absence of adolescent oligodendrogenesis. These results establish a critical role for oligodendrocytes in shaping the maturation and stabilization of cortical circuits and support the concept of developmental myelination acting as a functional brake on neuronal plasticity.

Relative to other cellular developmental processes in the brain, oligodendrogenesis and myelination occur in an extremely protracted manner, spanning the first few postnatal weeks in mice and the first three decades in humans. One theory for explaining this unique timing posits that developmental myelination stabilizes neuronal circuits and restricts the ability of cortical neurons to undergo certain forms of experience-dependent plasticity in adulthood[2–4,6]. The progression of myelination within the mouse visual cortex supports this idea, in which an accumulation of myelin within the input layer of the cortex coincides with the closure of a critical period for experience-induced functional neuronal plasticity[6]. Furthermore, sensory deprivation induces myelin remodelling within visual cortex[7], suggesting a potential link between sensory experience, myelination and neuronal function. To better define the relationship between visual experience and oligodendrocyte dynamics, we used genetic lineage tracing to track the generation of mature oligodendrocytes in the adolescent visual cortex following visual deprivation. For direct testing of the hypothesis that developmental myelination restricts neuronal plasticity, we genetically inhibited the generation of mature oligodendrocytes and progression of myelination during adolescence and assessed functional and structural experience-induced neuronal plasticity in the adult visual cortex. These experiments address a fundamental, as yet untested, hypothesis about brain development and inform our broader understanding of how

oligodendrocytes and myelin influence neuronal circuit function and plasticity.

## Sensory experience modulates oligodendrocyte lineage dynamics in the adolescent visual cortex

Mature, myelinating oligodendrocytes arise from oligodendrocyte precursor cells (OPCs). By crossing a mouse line that specifically expresses an inducible Cre in OPCs (NG2CreER) with a Cre-dependent reporter (tau-membrane-bound green fluorescent protein (mGFP)), we can identify newly generated oligodendrocytes by mGFP expression[8,9] because only cells that are OPCs at the time of tamoxifen administration will recombine the reporter. To determine whether sensory experience during adolescence can influence oligodendrocyte maturation, we gave tamoxifen to 4-week-old NG2CreER:tau-mGFP mice and used eyelid suturing to monocularly deprive them for 10 days, then quantified oligodendrogenesis in the visual cortex (Fig. 1a). OPCs that recombined the reporter were detected by coexpression of mGFP and the OPC-specific protein PDGFRα, whereas new premyelinating oligodendrocytes expressed mGFP but not PDGFRα and new mature oligodendrocytes expressed mGFP and the myelin protein MBP (Fig. 1b and Extended Data Fig. 1a). These newly formed mGFP⁺MBP⁺ myelin sheaths were flanked by CASPR immunoreactivity (Fig. 1f), an axonal protein clustered at paranodes that form between axons

[1]Department of Neurology, Weill Institute for Neurosciences, University of California San Francisco, San Francisco, CA, USA. [2]Department of Physiology, Kavli Institute for Fundamental Neuroscience and Weill Institute for Neurosciences, University of California San Francisco, San Francisco, CA, USA. [3]Departments of Neurosurgery and Neurology and Neurological Sciences, Stanford University, Stanford, CA, USA. ✉e-mail: wen.xin@ucsf.edu; jonah.chan@ucsf.edu

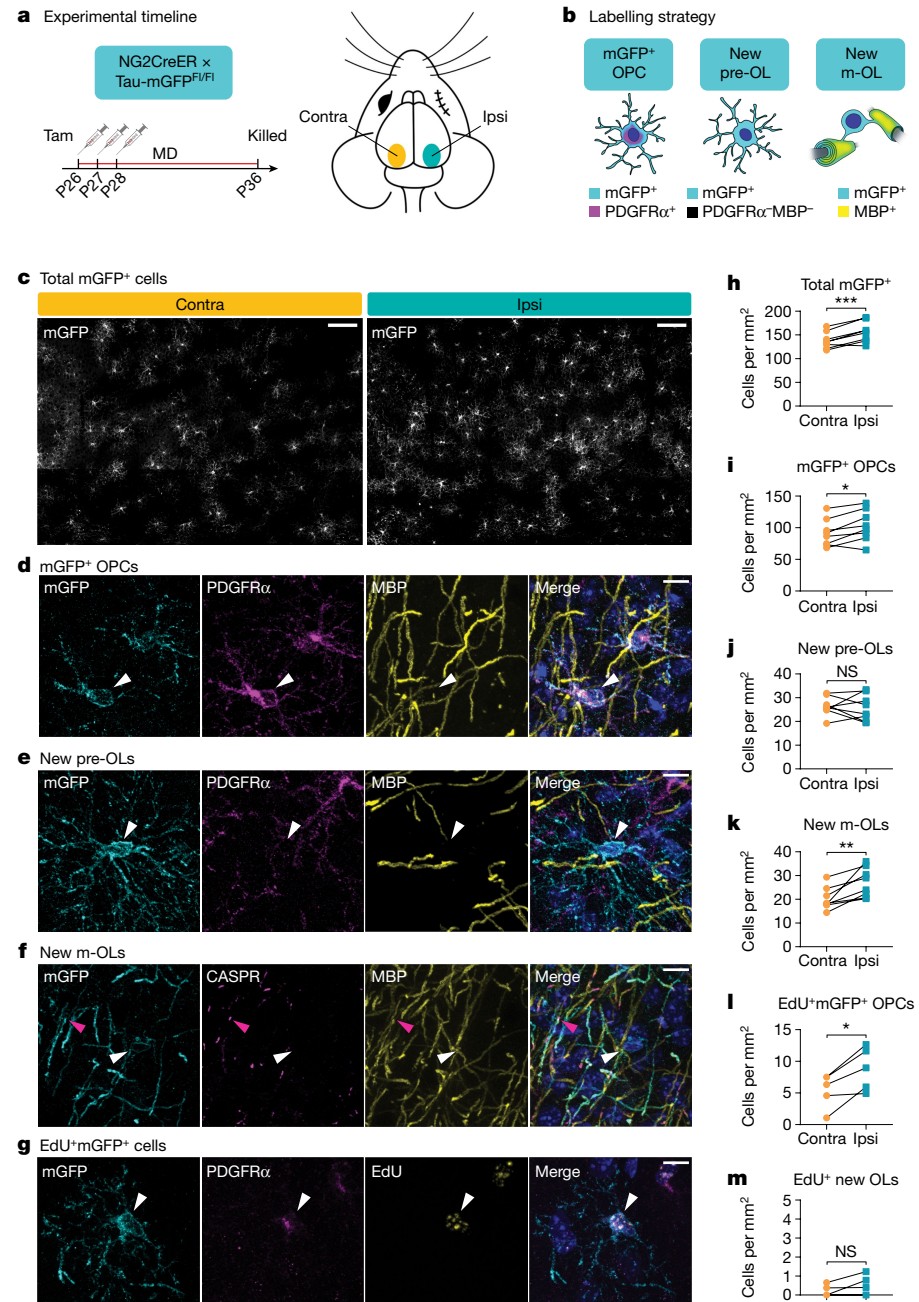

**Fig. 1 | Sensory experience during adolescence modulates oligodendroglial dynamics. a**, Experimental strategy and timeline. **b**, Labelling strategy for identification of OPCs, newly formed premyelinating oligodendrocytes (pre-OLs) and mature oligodendrocytes (m-OLs). **c,h**, Representative images (**c**) and quantification (**h**) of mGFP[+] cells from both hemispheres of visual cortex. Two-tailed paired $t$-test, $n = 8$ mice, $P = 0.0009$. **d,i**, Example image (**d**) and quantification (**i**) of mGFP[+] OPCs. Two-tailed paired $t$-test, $n = 8$ mice, $P = 0.0183$. **e,j**, Example image (**e**) and quantification (**j**) of newly formed pre-OLs. Two-tailed paired $t$-test, $n = 8$ mice, $P = 0.7584$. **f,k**, Example image (**f**) and quantification (**k**) of newly formed m-OLs. Two-tailed paired $t$-test, $n = 8$ mice, $P = 0.0064$. **g,l**, Example image (**g**) and quantification (**l**) of mGFP[+]EdU[+] OPCs. Two-tailed paired $t$-test, $n = 5$ mice, $P = 0.018$. **m**, Quantification of mGFP[+]EdU[+]PDGFRα[−] newly formed OLs. Two-tailed paired $t$-test, $n = 5$ mice, $P = 0.1763$. White arrowheads represent cell bodies, pink arrowheads MBP[+]CASPR[+]mGFP[+] myelin sheath. *$P < 0.05$, **$P < 0.01$, ***$P < 0.001$. Scale bars, 100 μm (**c**), 10 μm (**d**–**g**). Additional statistical details available in Supplementary Table 1. Contra, hemisphere contralateral to the deprived eye; ipsi, hemisphere ipsilateral to the deprived eye; MD, monocular deprivation; NS, not significant; tam, tamoxifen.

and the ends of compact myelin sheaths[10], which indicates that these sheaths are mature and functional myelin internodes. Overall, the contralateral (contra) cortex—which primarily receives visual inputs from the deprived eye—contained fewer mGFP[+] cells than the ipsilateral (ipsi) cortex (Fig. 1c,h). By lineage stage, there were slightly fewer mGFP[+] OPCs, and also fewer PDGFRα[+] OPCs overall, in the contralateral cortex (Fig. 1d,i and Extended Data Fig. 1b). The percentage of recombined OPCs was similar in both hemispheres (Extended Data Fig. 1c). There was no difference in the number of new premyelinating oligodendrocytes between the two hemispheres (Fig. 1e,j) but a significant decrease in the number of mature myelinating oligodendrocytes in the deprived cortex (Fig. 1f,k). Thus, sensory deprivation during adolescence modulates oligodendroglial dynamics in a stage-dependent manner.

There are two potential explanations for the decrease in mature oligodendrocytes in the contralateral cortex. Sensory deprivation may modulate OPC proliferation, which changes the density of OPCs and, indirectly, the density of mature oligodendrocytes that would be generated. Alternatively, sensory deprivation may directly alter oligodendrocyte survival and myelination. To distinguish between these two possibilities, we administered the thymidine analogue 5-ethynyl-2′-deoxyuridine (EdU), which can be incorporated into the newly synthesized DNA of dividing cells, at the same time as tamoxifen (Extended Data Fig. 1e). Following 10 days of monocular deprivation, there were fewer EdU$^+$ cells overall (Extended Data Fig. 1d,f) and fewer EdU$^+$ OPCs in the contralateral cortex, indicating that fewer OPCs had proliferated in the contralateral cortex during the experimental window (Fig. 1g,l). However, the number of newly generated oligodendrocytes that arose from proliferated OPCs (mGFP$^+$EdU$^+$PDGFRα$^-$) was very low in both hemispheres—that is, between zero and one cell per square millimetre (Fig. 1m)—compared with roughly 100 overall newly generated oligodendrocytes (Fig. 1i). Thus, the majority of new oligodendrocytes generated during the 10 days of monocular deprivation arose from OPCs that did not first proliferate, consistent with previous studies that used in vivo imaging to track oligodendrocyte dynamics in the cortex[11,12]. As such, an indirect effect of altered proliferation is unlikely to account for the difference in mature oligodendrocyte density between contralateral and ipsilateral cortex. Taken together, these results suggest that sensory deprivation during adolescence alters the rate of oligodendrocyte survival and myelination in the visual cortex.

## OPC-specific MYRF deletion in adolescent mice prevents oligodendrogenesis and myelination

Having established that sensory experience regulates oligodendrocyte maturation during adolescence, we next asked whether blocking adolescent oligodendrogenesis could alter neuronal function and plasticity within the visual cortex. We generated *Pdgfra-creER:Myrf*$^{Fl/Fl}$ mice, which enables OPC-specific deletion of MYRF, a transcription factor that is necessary for oligodendrocyte differentiation[13]. To prevent adolescent oligodendrogenesis, we administered tamoxifen to CreER$^+$ (cKO) and CreER$^-$ (control, CTL) mouse pups from postnatal day (P) 10 to P14 (Fig. 2a). In CTL mice, there was a rapid accumulation of ASPA$^+$ mature oligodendrocytes from 4 to 8 weeks of age, whereas the number of oligodendrocytes in cKO mice plateaued at 4 weeks of age (Fig. 2b,e and Extended Data Fig. 2). By 8 weeks of age, there was a pronounced stage-dependent decrease in the number of oligodendrocytes (Extended Data Fig. 3). Accordingly, the pattern of myelination in visual cortex remained sparse and patchy in 8-week-old cKO mice (Fig. 2d and Extended Data Fig. 2). Furthermore, the numbers of CASPR$^+$ nodes and heminodes, structures associated with functional, compact myelin sheaths[10], were significantly reduced in MYRF cKO mice (Extended Data Fig. 4). Given the global nature of our genetic manipulation, we also assessed oligodendrocyte density in earlier stages of the visual pathway—that is, optic nerve and lateral geniculate nucleus—at 8 weeks. Optic nerve oligodendrocyte density was unchanged and myelination grossly intact in MYRF cKO mice (Extended Data Fig. 5a,b). In the lateral geniculate nucleus we did observe a decrease in the number of oligodendrocytes in cKO mice, although this decrease was smaller and overall patterns of myelination were less affected than in visual cortex (Extended Data Figs. 2 and 5c,d). The heterogeneous effect of MYRF deletion along the visual pathway is probably driven by differences in the timing of developmental oligodendrogenesis, wherein oligodendrocytes in proximal portions of the visual pathway such as optic nerve differentiate much earlier[14] and are therefore not affected by subsequent MYRF deletion.

MYRF deletion did not alter the density of OPCs in either adolescent or adult visual cortex (Fig. 2c,f and Extended Data Fig. 3a), probably due to the exquisite ability of OPCs to maintain homeostatic density in vivo[11].

However, blocking differentiation could push OPCs into continuous proliferation instead, with homeostatic density being maintained by a simultaneous increase in cell death. To investigate this possibility, we injected adult MYRF cKO and littermate control mice with EdU for 5 days to assess the rate of OPC proliferation in visual cortex. Similar to our observations with sensory deprivation, decreasing differentiation—in this case by genetic manipulation—was associated with a decrease in OPC proliferation (Extended Data Fig. 5e–g). We also did not detect any signs of widespread cell death by either TUNEL staining or cleaved caspase-3 immunoreactivity (Extended Data Fig. 6a–d), signs of overt gliosis as indicated by glial fibrillary acidic protein immunoreactivity (Extended Data Fig. 6e) or any changes in the density of astrocytes (Extended Data Fig. 6f,g) or microglia (Extended Data Fig. 6h,i). Thus, MYRF deletion potently inhibits oligodendrogenesis and myelination in visual cortex without induction of widespread cell death or gross alteration of the density and morphology of other glial populations.

## Adolescent oligodendrogenesis enhances adult visual cortex activity and limits experience-dependent neuronal plasticity

Myelination has long been hypothesized to be important in the regulation of cortical neuronal maturation and plasticity[2,15,16], but the functional consequence of preventing developmental oligodendrogenesis and myelination has not been directly tested. We first verified that the loss of adolescent oligodendrogenesis does not induce neurodegeneration in visual cortex by immunostaining with a neurofilament light-chain (NF-L) antibody that binds an epitope of NF-L accessible only in degenerating axons[17]. As a positive control, we detected axonal degeneration in the spinal cord of mice that underwent experimental autoimmune encephalitis, particularly in regions with myelin loss (Extended Data Fig. 7a). By contrast, we saw little to no degeneration in the adult visual cortex of control and MYRF cKO mice (Extended Data Fig. 7b). Thus, unlike in demyelination, a lack of developmental myelination does not seem to be associated with axonal degeneration.

For assessment of functional neuronal activity in the visual cortex, we used intrinsic signal optical imaging, which enables non-invasive longitudinal monitoring of bulk neuronal activity during visual stimulation[18]. Adult mice were lightly anaesthetized and head fixed, and a visual stimulus was delivered to the ipsilateral or contralateral eye in the binocular portion of the mouse visual field (Fig. 2g). MYRF cKO mice exhibited normal retinotopic organization in the visual cortex (Extended Data Fig. 8a), as expected, given that cortical retinotopy is established by P15 (ref. 19) and oligodendrocyte density remains comparable between control and MYRF cKO mice before P28 (Fig. 2b,e). However, visual cortex responses to visual stimulation of either eye were significantly weaker in MYRF cKO mice compared with control mice, with a more pronounced decrease in ipsilateral responsiveness (Fig. 2i). As a result, baseline ocular dominance—that is, the relative strength of visual cortex responses to stimulation of the contralateral versus ipsilateral eye—was significantly higher in MYRF cKO mice (Fig. 2l). These results indicate that adolescent oligodendrogenesis and myelination are required for proper visual cortex maturation and function in adulthood.

It has previously been proposed that myelin in the visual cortex may gate the ability of neurons to undergo experience-dependent plasticity[2,4,6,15]. However, no study to date has functionally tested this hypothesis by cell type-specific manipulation of oligodendrocyte maturation or myelination. Therefore, we examined experience-dependent plasticity in the adult visual cortex of control and MYRF cKO mice. A hallmark of critical period experience-dependent plasticity is the loss of visual cortex responsiveness to the deprived eye following a brief period of monocular deprivation[5,20,21]. In wild-type adult mice, monocular deprivation induces an increase in visual cortex response to the non-deprived eye but no change in response to the deprived

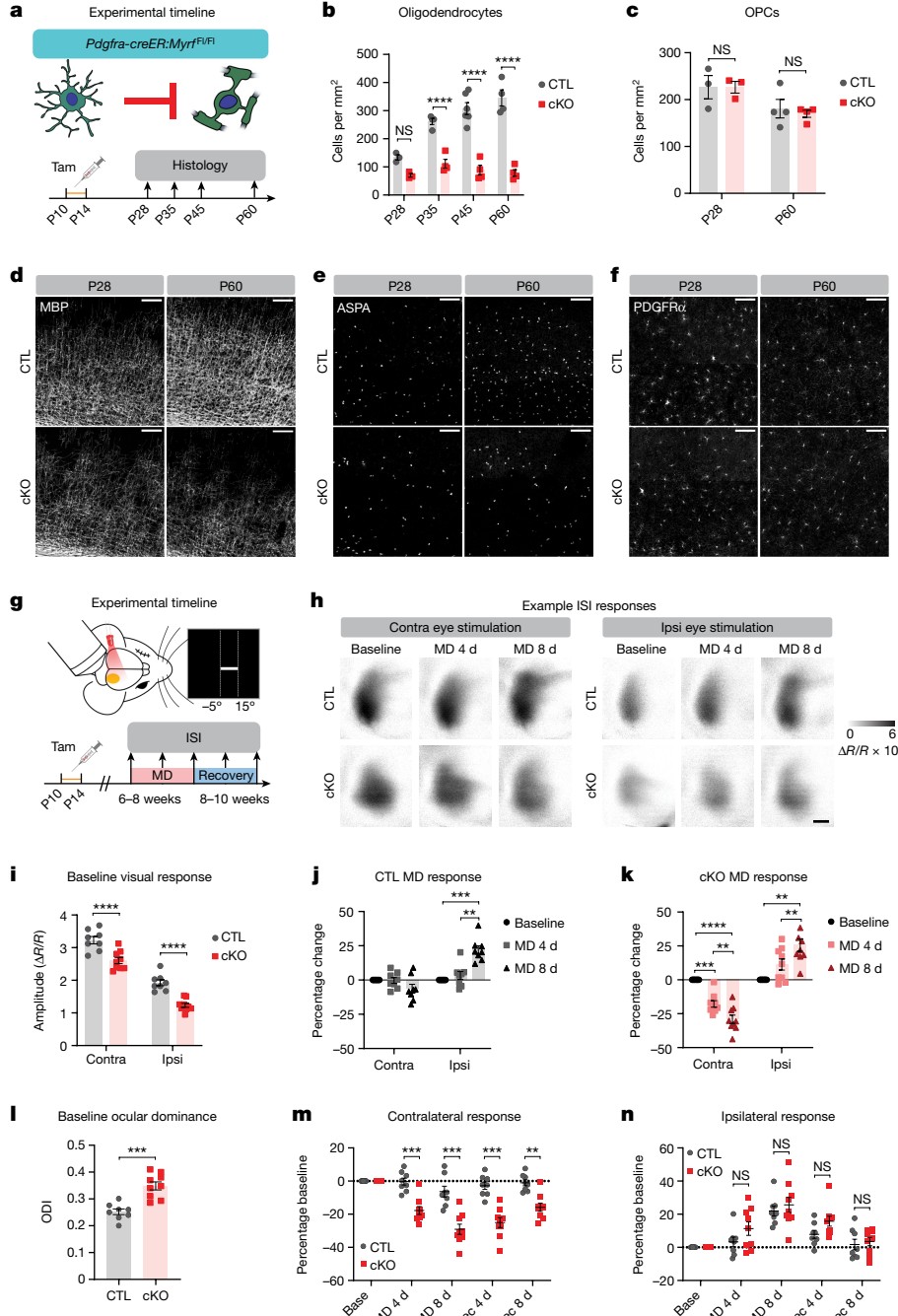

**Fig. 2 | Impairment of adolescent oligodendrogenesis disrupts adult visual cortex activity and enhances experience-dependent neuronal plasticity. a**, Experimental strategy and histology timeline. **b,e**, Quantification (**b**, $P = 0.0002$) and example images (**e**) of mature oligodendrocytes in the visual cortex at P28 and P60 in CreER⁻ (CTL) and CreER⁺ (cKO) mice. **c,f**, Quantification (**c**, $P = 0.7504$) and example images (**f**) of OPCs in visual cortex. **d**, Example images of myelin in visual cortex. **g**, Experimental strategy and timeline for intrinsic signal imaging (ISI). **h**, Example images from ISI. **i,l**, Amplitude of ISI responses (**i**, $P < 0.0001$) and ocular dominance (**l**) in binocular visual cortex at baseline ($P = 0.0001$). **j**, Change in ISI responses in adult control mice following 4 or 8 days of MD (contra, $P = 0.1804$; ipsi, $P < 0.0001$). **k**, Change in ISI responses in adult cKO mice (contra, $P < 0.0001$; ipsi, $P < 0.0001$). **m**, Change in ISI

responses to contralateral eye stimulation following MD and recovery (rec) ($P < 0.0001$). **n**, Change in ISI responses to ipsilateral eye stimulation ($P = 0.3008$). **b,c**, Two-way analysis of variance (ANOVA) with Sidak's multiple-comparisons test, $n = 3–6$ mice per age, per genotype; **i**, two-way ANOVA with Sidak's multiple-comparisons test; **j,k**, one-way repeated-measures ANOVA followed by Tukey's multiple-comparisons test; **l**, unpaired two-tailed $t$-test. **m,n**, two-way ANOVA with Sidak's multiple-comparisons test; **i–n**, $n = 8$ CTL mice and $n = 9$ cKO mice. Data presented as mean ± s.e.m. **$**P < 0.01$, ***$P < 0.001$, ****$P < 0.0001$. Additional statistical details are provided in Supplementary Table 1. Scale bars, 100 μm (**d–f**), 0.5 mm (**h**). MD 4 d/8 d, following 4 days/8 days of MD; ODI, ocular dominance index, defined as (contra − ipsi responses)/(contra + ipsi). $\Delta R/R$, change in relative reflectance.

eye[20,21]. Indeed, in adult control mice we observed stable visual cortex responses to the deprived eye following 4 or 8 days of monocular deprivation, although visual cortex responses to the non-deprived eye increased (Fig. 2h,j and Extended Data Fig. 8b), consistent with

previous studies[20,22,23]. By contrast, adult MYRF cKO mice exhibited pronounced decreases in visual cortex responsiveness to the deprived eye following monocular deprivation (Fig. 2h,k and Extended Data Fig. 8c). Comparing control and MYRF cKO visual cortex responses, we found

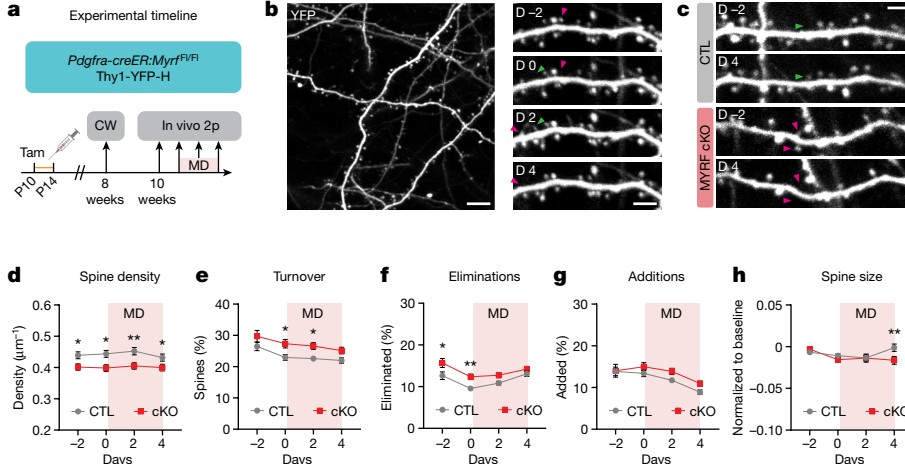

**Fig. 3 | Adult mice with impaired adolescent oligodendrogenesis have fewer spines and higher spine turnover. a**, Experimental strategy and timeline. **b**, Left, example two-photon (2p) image from an adult mouse visual cortex, contralateral to the deprived eye (z-projection of a 7-μm-thick volume). Right, example dendrite imaged at 2 days pre-MD (D −2), just before MD (D 0), 2 days following MD (D 2) and 4 days following MD (D 4) (z-projection of a 5-μm-thick volume). **c**, Example dendrites from control and cKO mice (z-projections of 3-μm-thick volumes). **d–h**, Average spine density (**d**, P = 0.0071), spine turnover (**e**, P = 0.0033), spine eliminations (**f**, P = 0.0002), spine additions (**g**, P = 0.0634) and spine size (**h**, P = 0.1175, interaction P = 0.0486) in CTL and cKO visual cortex dendrites. Green arrowheads denote spines that were added, pink arrowheads spines that were eliminated. **d,h**, Two-way repeated-measures ANOVA followed by Holm−Sidak multiple-comparisons test, n = 111 dendrites from ten CTL mice and n = 97 dendrites from ten cKO mice. **e–g**, Mixed-effects analysis (restricted maximum likelihood) followed by Holm−Sidak multiple-comparisons test, n = 111 dendrites from ten CTL mice and n = 97 dendrites from ten cKO mice. Data presented as mean ± s.e.m. *P < 0.05, **P < 0.01. Scale bars, 5 μm (**b** (right), **c**) and 10 μm (**b**, left). Additional statistical details are provided in Supplementary Table 1. CW, cranial window.

that contralateral responses following monocular deprivation were significantly different between groups (Fig. 2m) whereas ipsilateral responses were similar (Fig. 2n). These results demonstrate a key role for adolescent oligodendrogenesis and myelin in limiting functional experience-dependent neuronal plasticity in the visual cortex.

## Adolescent oligodendrogenesis regulates structural synaptic plasticity in adult visual cortex

A key anatomical correlate for functional experience-dependent neuronal plasticity is the physical rewiring of visual cortex circuitry[24–27]. In addition to functional changes in visual cortex neuronal activity, monocular deprivation during adolescence induces the elimination of dendritic spines in visual cortex pyramidal neurons[24–26]. To test whether adolescent oligodendrogenesis and myelination can also regulate structural neuronal plasticity, we crossed MYRF cKO mice with Thy1-yellow fluorescent protein (YFP)-H mice, which allowed us to visualize a subset of pyramidal neurons and their dendritic spines in the visual cortex (Extended Data Fig. 9a). We performed in vivo two-photon longitudinal imaging in adult mice to track dendritic spines in the contralateral cortex over a period of normal vision and monocular deprivation (Fig. 3a,b). Overall, spine density was lower in MYRF cKO mice than in littermate control mice (Fig. 3c,d), echoing the decrease in functional visual cortex responses we observed in cKO mice (Fig. 2i). The relative decrease in spine density was maintained throughout the period of monocular deprivation (Fig. 3d). Spine turnover, on the other hand, was higher in MYRF cKO mice (Fig. 3e), which was driven by an increase in spine eliminations (Fig. 3f) as well as a trend towards increased spine additions (Fig. 3g).

Spines that were added or eliminated during the imaging sessions accounted for approximately one-third of the total population (Fig. 3e), meaning most spines remained present throughout the experiment. However, persistent spines can still exhibit functionally significant structural plasticity; indeed, spine size is tightly correlated with synaptic strength[28,29]. Overall, we observed a difference in spine size changes following 4 days of monocular deprivation, with cKO mice exhibiting

a decrease in relative spine size compared with control mice (Fig. 3h), as well as a negative (leftwards) shift in the distribution of spine size changes (Extended Data Fig. 9b), indicating that more spines had decreased in size with monocular deprivation. For both groups, the size change of each spine following 2 days of monocular deprivation was strongly associated with that following 4 days of monocular deprivation, meaning that spines which had increased or decreased after 2 days had continued to increase or decrease after 4 days (Extended Data Fig. 9c,d).

Previous studies examining experience-induced spine plasticity have found that spine morphology changes tend to be spatially clustered along segments of dendrites[30,31]. This spatial clustering leads to supralinear summation, both electrically and biochemically, in the dendritic integration of individual changes in synaptic strength[32–35], meaning that a similar level of spine changes can have stronger or weaker effects on circuit activity depending on the extent of spatial coordination. Given the practical implications of spatially clustered spine changes on circuit function, we performed a nearest-neighbour analysis to assess potential clustering of spine plasticity within control and cKO mice, in which the directional change of each spine was compared with that of its nearest spine neighbour (Fig. 4a). In control mice, there was no correlation between the direction of spine size change of nearest neighbours (Fig. 4b and Extended Data Fig. 9e). By contrast, there was a significant positive correlation between the size changes of nearest spine neighbours in cKO mice (Fig. 4b and Extended Data Fig. 9f). At the level of dendrites, the percentage of spine pairs that increased together was similar between groups (Fig. 4c), but cKO dendrites had a higher percentage of spine pairs that decreased together (Fig. 4d). Combined, cKO dendrites had more spine pairs changing in the same direction than control dendrites (Fig. 4e), and a similar percentage of spine pairs changing in the opposite direction (Fig. 4f). These results indicate that spine changes induced by monocular deprivation are more spatially clustered in cKO mice.

Given that we observed a greater number of spines decreasing in size in cKO mice (Fig. 3h and Extended Data Fig. 9b), it is possible that the higher level of spine change clustering in cKO mice is simply due

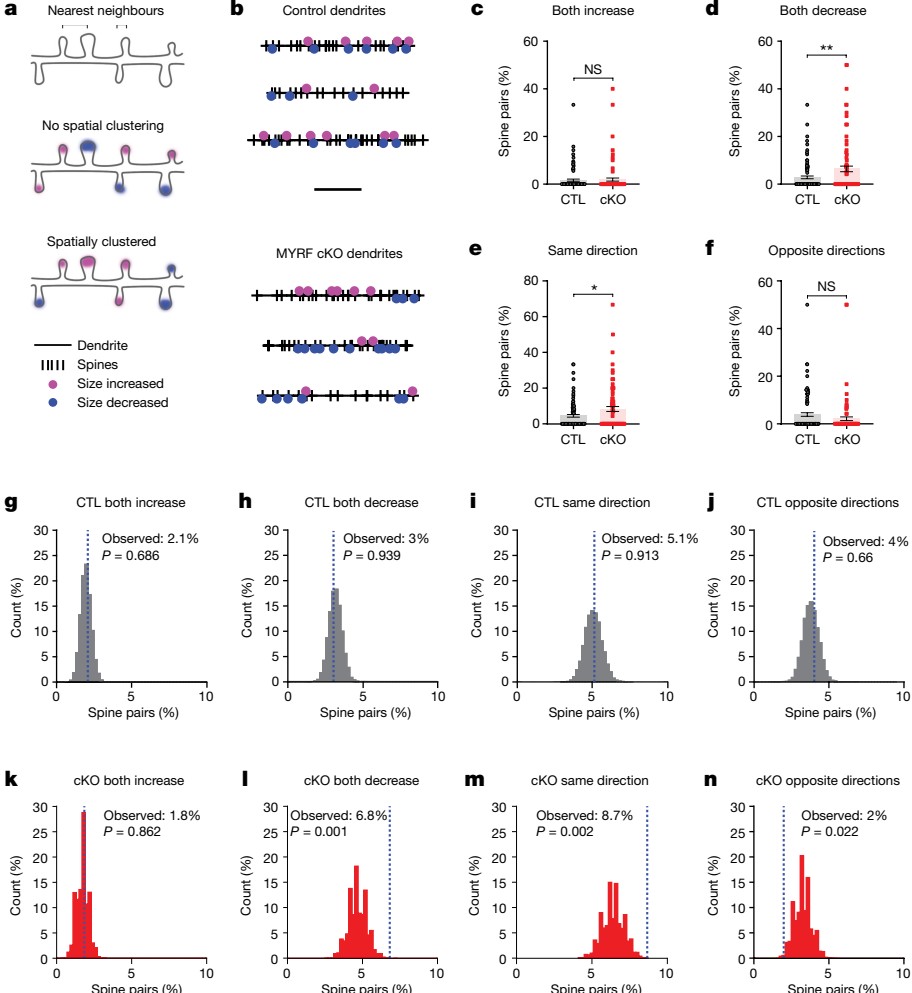

**Fig. 4 | Monocular deprivation induces spatially clustered spine size decreases in adult mice with impaired adolescent oligodendrogenesis.**
**a**, Schematic of nearest-neighbour analysis. Brackets indicate two examples of pairs of nearest-neighbouring spines. **b**, Spinodendrograms of three example dendrites per group. Scale bar, 20 μm. **c**–**f**, Percentage of spine pairs per dendrite that either increased (**c**, $P = 0.7427$), decreased (**d**, $P = 0.0052$), changed in the same direction (**e**, $P = 0.0112$) or changed in opposite directions (**f**, $P = 0.1052$) in CTL and cKO mice. **g**, Distribution of percentage of spine pairs that increased together ($P = 0.686$) from 10,000 random spine–location pairings in control mice; blue dashed line denotes the observed percentage of spine pairs. **h**–**j**, Distribution of percentages of random spine pairs that either decreased together (**h**, $P = 0.939$), changed in the same direction (**i**, $P = 0.913$) or changed in opposite directions (**j**, $P = 0.66$) in control mice. **k**–**n**, Distribution of percentages of random spine pairs (and observed percentage, blue dashed line) that either increased (**k**, $P = 0.862$), decreased (**l**, $P = 0.001$), changed in the same direction (**m**, $P = 0.002$) or changed in opposite directions (**n**, $P = 0.022$) in cKO mice. **c**–**f**, Unpaired two-tailed $t$-tests, $n = 110$ dendrites from ten CTL mice and $n = 96$ dendrites from ten cKO mice; data presented as mean ± s.e.m. **g**–**n**, Monte Carlo $P$ values, $n = 996$ spine pairs from ten CTL mice and $n = 704$ spine pairs from ten cKO mice. *$P < 0.05$, **$P < 0.01$. Additional statistical details are provided in Supplementary Table 1.

to more spine decreases present along each dendrite. To determine whether this was the case, we compared the observed spine pairs with a distribution of spine pairs generated by randomly shuffling the changes in spine size along all spine positions in each dendrite (Extended Data Fig. 9g,h). In control mice, the observed spine pairs increasing or decreasing together fell within the distribution of random spine pairs, indicating that the level of spine clustering we observed was occurring at chance levels (Fig. 4g–j). In cKO mice, the percentage of spine pairs increasing together fell within the distribution of random spine pairs (Fig. 4k), but the percentage of spine pairs decreasing together was significantly higher than would be predicted by chance (Fig. 4l). Altogether, we observed many more spine pairs changing in the same direction (Fig. 4m), and fewer spine pairs changing in the opposite direction (Fig. 4n), in cKO mice than expected based on the overall number of spines increasing and decreasing in size. Thus, cKO mice exhibit an enhanced level of spatially clustered decreases in spine size following monocular deprivation. These differences in structural

plasticity may underlie the selective decrease in functional visual cortex responses we observed following monocular deprivation in adult cKO mice.

## Inhibitory synaptic transmission in adult visual cortex is impaired in the absence of adolescent oligodendrogenesis

Increased inhibition—specifically, the maturation of parvalbumin-expressing interneurons—is a key regulator of visual cortex experience-dependent plasticity at the circuit level[36]. Disruption of perineuronal net formation around parvalbumin neurons or transplantation of immature inhibitory neurons into the visual cortex results in increased visual cortex plasticity[37,38]. Parvalbumin neurons are also heavily myelinated in the mammalian cortex[39,40]. We therefore hypothesized that inhibition may be impaired in the absence of adolescent oligodendrogenesis. In adult visual cortex,

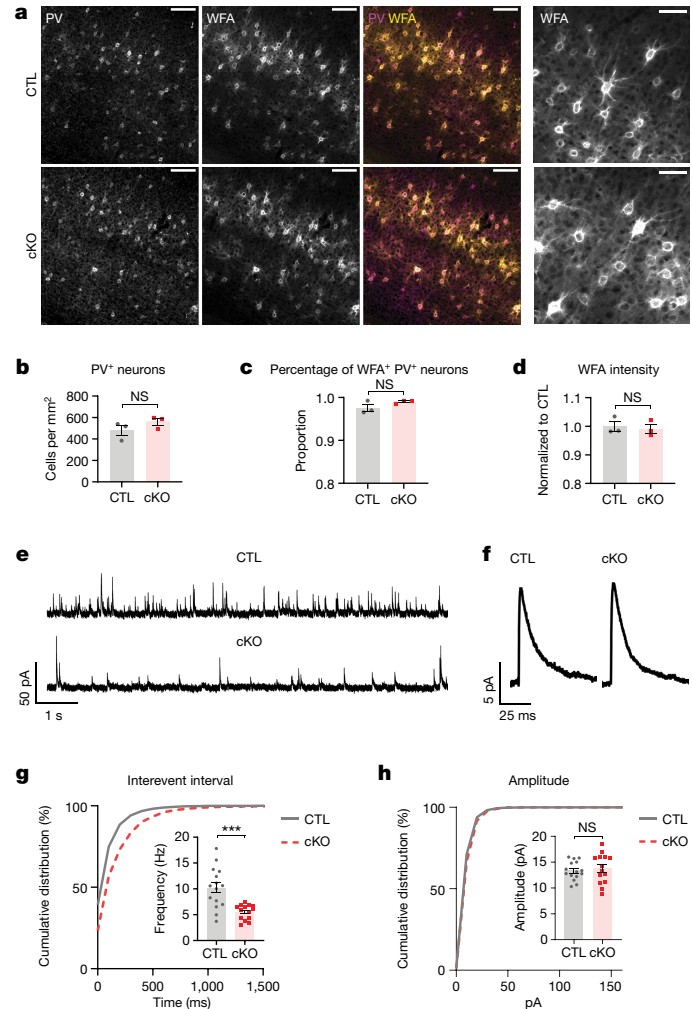

**Fig. 5 | Inhibitory synaptic transmission is impaired in adult mice lacking adolescent oligodendrogenesis. a**, Example images of parvalbumin (PV) and WFA immunostaining in the adult visual cortex of CreER⁻ (CTL) and CreER⁺ (cKO) mice. **b–d**, Quantification of PV⁺ neuron density (**b**, $P = 0.2538$), percentage of PV⁺ neurons with WFA⁺ perineuronal nets (**c**, $P = 0.1773$) and WFA intensity in visual cortex (**d**, $P = 0.718$). **e**, Example voltage-clamp traces recorded at 0 mV from layer V visual cortex pyramidal neurons in the presence of tetrodotoxin, D-(-)-2-amino-5-phosphonopentanoic acid (APV) and 2,3-dioxo-6-nitro-1,2,3,4-tetrahydrobenzo[f]quinoxaline-7-sulfonamide (NBQX). **f**, Example waveforms of mIPSCs (average trace obtained from all events recorded from one cell for each genotype). **g**, Cumulative distribution of mIPSC interevent interval. Inset, quantification of mIPSC frequency ($P = 0.0003$). **h**, Cumulative distribution of mIPSC amplitude. Inset, quantification of mIPSC amplitude ($P = 0.6413$). **b–d**, Unpaired two-tailed *t*-test, $n = 3$ CTL mice and $n = 3$ cKO mice; **g–h**, unpaired two-tailed *t*-test, $n = 15$ cells from five CTL mice and $n = 14$ cells from five cKO mice. Data presented as mean ± s.e.m. \*\*\*$P < 0.001$. Additional statistical details are provided in Supplementary in Table 1.

MYRF cKO mice had comparable numbers of parvalbumin neurons to control mice (Fig. 5a,b) and the presence and intensity of perineuronal nets—visualized by *Wisteria floribunda* agglutinin (WFA) immunostaining—were similarly preserved (Fig. 5a,c,d). However, the frequency of miniature inhibitory currents (mIPSCs) onto visual cortex pyramidal neurons was significantly reduced in adult MYRF cKO mice (Fig. 5e,g), unaccompanied by any change in mIPSC amplitude or major differences in mIPSC kinetics (Fig. 5f,h and Extended Data Fig. 10), probably reflecting a presynaptic change in inhibitory transmission. Thus, adolescent oligodendrogenesis and myelination are required for proper inhibitory signalling in the adult visual

cortex, the absence of which may contribute to enhanced structural and functional neuronal plasticity in adulthood.

## Discussion

In the present study, using cell type-specific genetic approaches, we found that sensory deprivation during adolescence reduces oligodendrocyte maturation and that blocking of adolescent oligodendrogenesis enhances both functional and structural neuronal plasticity in the adult visual cortex. In addition to enhanced plasticity following sensory deprivation, we observed a significant reduction in inhibitory synaptic transmission in the adult cortex of mice lacking adolescent oligodendrogenesis, providing a potential circuit-level basis for enhanced neuronal plasticity. Furthermore, visual cortex responsiveness to visual stimuli was strongly reduced and visual cortex responses were dominated by the contralateral eye, reminiscent of an immature cortical state[41]. Our results provide empirical evidence for the hypothesis that developmental myelination is required for proper circuit maturation and functionally limits experience-dependent neuronal plasticity.

The structural and functional neuronal changes we observed in mice lacking adolescent oligodendrogenesis were not accompanied by any signs of gliosis, neurodegeneration or widespread cell death. Thus, a lack of adolescent oligodendrogenesis and myelination does not appear to induce the same type of injury as genetic or inflammatory models of demyelination. Accordingly, we also saw a decrease, rather than an increase, in OPC proliferation when we genetically prevented oligodendrocyte differentiation, consistent with previous observations of differentiation being a trigger for proliferation outside of injury settings[11,12]. It is possible that apoptosis may be slightly elevated over the course of adolescence in mice with decreased oligodendrogenesis, driven by a decreased rate of successful oligodendrocyte integration, and resolves by adulthood. However, this would be taking place against a high background of physiological oligodendrocyte apoptosis[42] that occurs as part of the normal oligodendrocyte maturation process and therefore is not likely to be a strong driver of the enhanced plasticity we observed in MYRF cKO mice.

Although our study focuses on the role of developmental myelination, the ability of oligodendrocytes and myelin to regulate neuronal plasticity may expand well beyond development. Recent studies have reported a requirement for adult oligodendrogenesis in multiple forms of learning and memory[43–47], but it is still unclear how oligodendrocyte or myelin plasticity modulates the underlying circuit activity to influence behaviour[43]. Based on our findings within a sensory circuit, it is plausible that new oligodendrocytes and myelin may serve to stabilize the formation of new synaptic connections following learning to enable long-term memory. This possibility is further supported by imaging studies in vivo reporting neuronal cell type-specific and activity-driven changes in myelin sheath following sensory manipulations or motor learning[7,48]. Our results illuminate the potential for oligodendrocytes and myelin to act as a toggle between neuronal circuit stability and reorganization for lifelong brain plasticity, and implicate myelin as a key driver of circuit dysfunction in neurodevelopmental disorders presenting with impaired oligodendrocyte maturation[43,49,50].

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

## Methods

### Animals

All mice were handled in accordance with, and all procedures approved by, the Institutional Animal Care and Use Committee of the University of California San Francisco. Mice were group housed (between two and five per cage) throughout all experiments and given food and water ad libitum on a 12/12 h light/dark cycle in a temperature-controlled (22–24 °C) and humidity-controlled (40–60%) environment. Housing conditions adhered to the standards maintained by University of California San Francisco Institutional Animal Care and Use Committee, which include standard-sized mouse cages, bedding, nestlet, gnawing block and Enviro-dry nesting material. No additional environmental enrichment was provided. Males and females were used for all experiments. For tracking of oligodendrogenesis during adolescence, NG2CreER:tau-mGFP mice[51,52] (Jax, nos. 008538 and 021162) received 100 mg kg$^{-1}$ tamoxifen (Sigma, catalogue no. T5648) by oral gavage from P26 to P28. A subset also received 80 mg kg$^{-1}$ EdU (Carbosynth, catalogue no. NE08701) by intraperitoneal injection at the same time. For blocking of adolescent oligodendrogenesis, *Pdgfra-creER:Myrf*$^{Fl/Fl}$ mice[13,53] (Jax, nos. 018280 and 010607) received 100 mg kg$^{-1}$ tamoxifen by oral gavage from P10 to P14. For visualization of dendritic spines in vivo, *Pdgfra-creER:Myrf*$^{Fl/Fl}$ mice were crossed with Thy1-YFP-H mice[54] (Jax, no. 003782). Experimenters were blinded to animal genotype throughout data acquisition and analysis.

### Immunohistochemistry

Mice were deeply anaesthetized with Avertin and perfused transcardially with 4% paraformaldehyde in 1× PBS. Brain tissue was isolated and postfixed in this solution overnight at 4 °C, then stored in 1× PBS with 0.1% NaAz. Brains were sucrose protected (30% in PBS) before flash-freezing and sectioning coronally (30 μm) on a sliding microtome. Free-floating sections were permeabilized/blocked with 0.2% Triton X-100 and 10% normal goat serum in 1× PBS for 1 h at room temperature. Sections were incubated with primary antibodies prepared in 0.2% Triton X-100 and 10% normal goat serum in 1× PBS at 4 °C overnight. Sections were incubated with secondary antibodies in 10% normal goat serum in 1× PBS for 2 h at room temperature. Primary antibodies and concentrations used are as follows: rabbit anti-ASPA (1:1,000), chicken anti-GFP (1:1,000), rat anti-MBP (1:200), rabbit anti-PDGFRα (1:200), rabbit anti-cleaved caspase-3 (1:200), mouse anti-glial fibrillary acidic protein (1:1,000), human anti-SOX9 (1:2,000), rabbit anti-IBA1 (1:1,000), mouse anti-NF-L Degenotag (1:1,000), rabbit anti-NF-H (1:1,000), mouse anti-PV (1:1,000), biotinylated WFA (1:400), rabbit anti-CASPR (1:600) and mouse anti-BCAS1 (1:300); additional details are listed in Supplementary Table 2. The primary antibodies above have been validated for use in immunohistochemistry in mouse tissue, in published literature and on the manufacturer's websites. Secondary antibodies used included the following: Alexa Fluor 488-, 594- or 647-conjugated secondary antibodies to rabbit, mouse, human, chicken, rat or streptavidin (1:1,000, all raised in goat; purchased from Thermo Fisher Scientific or Jackson ImmunoResearch); additional details are listed in Supplementary Table 2. Cell nuclei were labelled with DAPI (Vector Laboratories). TUNEL immunostaining was performed on fixed brain sections according to the manufacturer's instructions using the Abcam TUNEL Assay Kit−BrdU-Red (abcam, catalogue no. ab66110).

### Fixed-tissue imaging and analysis

Tiled *z*-stacks (with 2 μm steps) spanning either 30 μm sections of visual cortex and lateral geniculate nucleus or 20 μm sections of optic nerve were taken with a Zeiss Axio Imager Z1 with ApoTome attachment and Zeiss Zen 2 (blue edition, v.2.0.0.0) software, using a ×10 objective. For quantification, images were taken from two or three sections per mouse. Cell density was quantified manually using Cell Counter in Fiji. Experimenters were blinded to genotype throughout imaging acquisition and analysis.

### Slice electrophysiology

Mice aged 8–12 weeks were anaesthetized with isoflurane. Brains were quickly removed and placed in ice-cold artificial cerebrospinal fluid (ACSF) containing 125 mM NaCl, 2.5 mM KCl, 2 mM CaCl$_2$, 1.25 mM NaH$_2$PO$_4$, 1 mM MgCl$_2$, 25 mM NaHCO$_3$ and 15 mM D-glucose. ACSF was saturated with 95% O$_2$ and 5% CO$_2$. Osmolarity was adjusted to 300–305 mOsm. Coronal sections (300 μm) containing visual cortex were prepared in ice-cold ACSF using a vibrating-blade microtome (Leica VT1200). Slices were recovered for 20 min at 32 °C and then transferred to ACSF at room temperature. Following the recovery period, slices were moved to a submerged recording chamber perfused with ACSF at a rate of 2–3 ml min$^{-1}$ at 30–31 °C, and brain slices were recorded within 5 h of recovery. Voltage-clamp recordings of mIPSCs were made using glass pipettes of resistance 2–4 MΩ, filled with internal solution containing 126 mM CsMeSO$_3$, 8 mM NaCl, 10 mM HEPES, 2.9 mM QX-314, 8 mM Na2-phosphocreatine, 0.3 mM GTP-Na, 4 mM ATP-Mg, 0.1 mM CaCl$_2$ and 1 mM EGTA, pH 7.2–7.3, osmolarity 285–290 mOsm. Input resistance was monitored online during recordings; cells with access resistance greater than 20 MΩ were excluded from analysis. Recordings were made at 0 mV holding potential. mIPSCs were pharmacologically isolated with tetrodotoxin (1 μM), NBQX (10 μM) and APV (50 μM) in the bath. Between 200 and 300 events per cell were analysed using a threshold of 2× baseline noise. Recordings were obtained with a Multiclamp 700B amplifier (Molecular Devices) using WinWCP software (University of Strathclyde, UK). Signals were filtered at 2 kHz, digitized at 10 kHz (NI PCIe-6259, National Instruments) and analysed offline using the MiniAnalysis Program (Synaptosoft). The experimenter was blinded to the genotype of the animals throughout recording and analysis.

### Monocular deprivation

Mice were anaesthetized using 5% isoflurane and anaesthesia was maintained with 2–3% isoflurane. The right eyelid was sutured closed by two mattress stitches, at either P26 for adolescent NG2CreER:tau-mGFP mice or 6–12 weeks for post-critical-period *Pdgfra-creER:Myrf*$^{Fl/Fl}$ mice. Meloxicam and buprenorphine were administered before and after surgery for pain management. Animals were checked daily to ensure that the sutured eye remained closed for the required duration of the experiment. Sutures were removed just before postmonocular deprivation ISI sessions. Eyes were flushed with sterile saline and checked for clarity under a microscope. Only mice without corneal opacities or signs of infection were used.

### ISI

Repeated optical imaging of intrinsic signals and quantification of ocular dominance were performed as previously described[18]. In brief, during recording, mice were anaesthetized with 0.7% isoflurane in oxygen applied via a home-made nose mask, supplemented with a single intramuscular injection of 20–25 μg chlorprothixene. Mice underwent a non-invasive procedure in which a headplate was fixed to the surface of the skull to enable head-fixed imaging, and images were recorded transcranially. Intrinsic signal images were obtained with a Dalsa 1M30 CCD camera (Dalsa) fitted with a 135 × 50 mm tandem lens (Nikon) and red interference filter (610 ± 10 nm), using custom Linux software. Frames were acquired at a rate of 30 per second, temporally binned by four frames and stored as 512 × 512 pixel images following spatial binning of 1,024 × 1,024 camera pixels by 2 × 2 pixels. The visual stimulus for recording the binocular zone, presented on a 40 × 30 cm$^2$ monitor placed 25 cm in front of the mouse, consisted of 2°-wide bars that were presented between −5 and 15° on the stimulus monitor (0°, centre of the monitor aligned to centre of the mouse) and moved continuously and periodically upward or downward at a speed of 10° s$^{-1}$. The phase and amplitude

of cortical responses at the stimulus frequency were extracted by Fourier analysis as previously described[18]. Response amplitude was taken as an average of at least four measurements. Ocular dominance index was computed as previously described[18]. In brief, the binocularly responsive region of interest (ROI) was chosen based on the ipsilateral eye response map following smoothing by low-pass filtering, using a uniform kernel of 5 × 5 pixels and thresholding at 40% of peak response amplitude. Ocular dominance score $(C − I)/(C + I)$ was computed for each pixel in this ROI, in which $C$ and $I$ represent the magnitude of response to contralateral and ipsilateral eye stimulation, respectively, followed by calculation of ocular dominance index as the average of ocular dominance score for all responsive pixels. Experimenters were blinded to genotype throughout imaging and analysis.

## CW surgery
At the age of 8–12 weeks, a square $3 × 3 \, mm^2$ cranial window (no. 1 coverslip glass, Warner Instruments) was placed over the left hemisphere of the cortex contralateral to the deprived eye. Mice were anaesthetized using 5% isofluorane and anaesthesia was maintained with 2–3% isoflurane. A craniotomy matching the size of the coverslip was cut using no. 11 scalpel blades (Fine Science Tools) and the coverslip carefully placed on top of the dura within the craniotomy without excessive compression of the brain. The window was centred using stereotactic coordinates 2 mm lateral and 3 mm posterior from bregma for visual cortex. The window and skull were sealed using dental cement (C&B Metabond, Parkell). A custom-made metal head bar was attached to the skull during surgery for head-fixed imaging. Mice were allowed to recover for 2–3 weeks before two-photon imaging.

## In vivo longitudinal imaging
Longitudinal in vivo two-photon imaging was performed, as previously described[31], with *Pdgfra-creER:Myrf*[Fl/Fl] mice crossed with Thy1-YFP-H mice. Specifically, apical dendrites of cortical pyramidal neurons expressing YFP were imaged repeatedly 10–100 µm below the cortical surface through the cranial window in mice under isoflurane anaesthesia. Images were acquired using a Bergamo II two-photon microscope system with a resonant scanner (Thorlabs) and a ×16/0.8 numerical aperture water-immersion objective lens (Nikon), using ThorImage LS software. YFP was excited at 925 nm with a mode-locked, tunable, ultrafast laser (InSightX3, Spectra-Physics) with 15–100 mW of power delivered to the back-aperture of the objective. Image stacks were acquired at 1,024 × 1,024 pixels with a voxel size of 0.12 µm in $x$ and $y$ and a $z$-step of 1 µm. Imaging frames from resonant scanning were averaged during acquisition to achieve a pixel dwell time equivalent of 1 ns. Up to six imaging regions were acquired for each mouse. Representative images shown in the figures were created by making $z$-projections of three-dimensional stacks and were median filtered and contrast enhanced.

## Analysis of in vivo spine imaging
Dendritic spines were analysed using the custom software Map Manager (https://mapmanager.net) written in Igor Pro (WaveMetrics) as previously described[31,55]. Experimenters were blinded to genotype throughout imaging acquisition and analysis. For annotation, the dendritic shaft was first traced using a modified version of the 'Simple Neurite Tracer' plug-in provided in ImageJ. Spine positions along a dendritic segment were manually identified by the location of the spine tip in three-dimensional image stacks of all imaging sessions. For longitudinal analysis, spines were further tracked across time by comparison of images from different sessions and connecting persistent spines. Rates of spine addition and elimination were calculated as the number of newly added or eliminated spines on a given imaging session divided by the total number of spines of that dendritic segment

on the previous imaging session. The turnover ratio represents the sum of spine addition and elimination.

The fluorescence intensity of dendritic spines was used as a proxy for spine size, and therefore a three-dimensional ROI was defined for each spine, the dendritic shaft (4 µm stretch) adjacent to that spine and a nearby background region. For comparison of intensity values between imaging sessions, and to account for small variations in daily imaging conditions, spine signal intensity was normalized to the signal on the adjacent dendritic shaft following background subtraction. Each spine value was subsequently normalized to an average of the baseline imaging sessions by first subtracting the baseline value and then dividing over the sum of the baseline and respective imaging day value. This normalizes spine size change values between −1 and +1. All spine analysis was performed for each dendritic segment, averaged per genotype and is presented as the average of values from two adjacent imaging sessions (−3 and −2, −1 and 0, 1 and 2 and so on) to increase clarity.

For analysis of spine clustering, spines were classified as either increasing, decreasing or stable based on their average change in size on days 1–4 compared with baseline. The threshold for these categories was set based on the variability in control mice and was defined at baseline ± 1 s.d. of size changes (±0.14). Nearest-neighbour analysis was calculated by finding the closest neighbour of every spine along each dendritic segment. Each nearest-neighbour pair was included once only in the dataset and pairs were excluded if their distance was either below 1.0 µm (to avoid overlapping ROIs) or above 3.5 µm. The fractions of nearest-neighbour spine pairs in which both spines increase, both decrease or changes occur in the same direction or in opposite directions were quantified to compare the degree of clustering between genotypes.

To test the statistical significance of clustering, nearest-neighbour analysis was performed on a pool of randomized spines in which spine size change values were randomly shuffled along all spine positions in each dendrite. A Monte Carlo $P$ value was calculated by summing the tail of the histograms from 10,000 pools of randomized spine pairings in which the nearest-neighbour analysis resulted in spine pair fractions that exceeded the real observed value.

## Statistics and reproducibility
All graphed values are shown as mean ± s.e.m. Statistical details of experiments (statistical tests used, statistical values, exact $n$ values) are listed in Supplementary Table 1. The number of animals included in each experiment was based on standards established in the literature. Statistics were performed using GraphPad Prism. Statistical significance was defined as $P < 0.05$. Tests for normality and equal variances were used to determine the appropriate statistical test to use. All reported $t$-tests were two-tailed, with Welch's correction when group variances were significantly different. For experiments with more than two groups, one-way ANOVA was used; for experiments with more than one variable, two-way ANOVA was used; for experiments with repeated measurements from the same animals, two-way repeated-measures ANOVA was used. All representative images were selected from one of a minimum of three independently repeated experiments with similar results.

## Reporting summary
Further information on research design is available in the Nature Portfolio Reporting Summary linked to this article.

## Data availability
All data supporting the findings of this study are available within the paper and its Supplementary Information. Raw in vivo imaging datasets are available from the corresponding authors on request. Source data are provided with this paper.

## Code availability

MapManager software and code developed for longitudinal tracking of dendritic spines are available at https://mapmanager.net/. Additional code for post hoc spine analysis is available at Zenodo (https://zenodo.org/doi/10.5281/zenodo.12170661)[56]. Code used for intrinsic signal imaging and analysis of ocular dominance is available at GitHub (https://github.com/mpstryker/ISI).

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

**Acknowledgements** We thank all members of the Chan laboratory for their feedback on the manuscript, and R. Yang for technical assistance with slice electrophysiology. This work was supported by the National Institutes of Health/National Institute of Neurological Disorders and Stroke (grant nos. F32NS116214/K99NS131200 to W.X., K99NS130078 to R.H.R. and R01NS115746 to J.R.C.), the National Institutes of Health/National Institute for Mental Health (grant no. R01MH125515 to J.R.C.), the National Institutes of Health/National Eye Institute (grant no. R01EY02874 to M.P.S.), the Dr. Miriam and Sheldon G. Adelson Medical Research Foundation (APND grant no. A130141 to J.R.C.), the Brain & Behavior Research Foundation (NARSAD award to W.X.), the Canadian Institutes of Health Research (award no. 491626 to A.Z.) and the Rachleff Family Endowment.

**Author contributions** W.X. and J.R.C. conceived the experiments. W.X., M.K. and R.H.R. performed imaging experiments. W.X., A.Z. and S.N. performed histology experiments. W.X. performed slice electrophysiology experiments. W.X., M.K., R.H.R., A.Z. and S.N. analysed data. J.B.D., M.P.S. and J.R.C. provided resources and scientific guidance. W.X. wrote the manuscript with input from all authors.

**Competing interests** The authors declare no competing interests.

**Additional information**
**Correspondence and requests for materials** should be addressed to Wendy Xin or Jonah R. Chan.

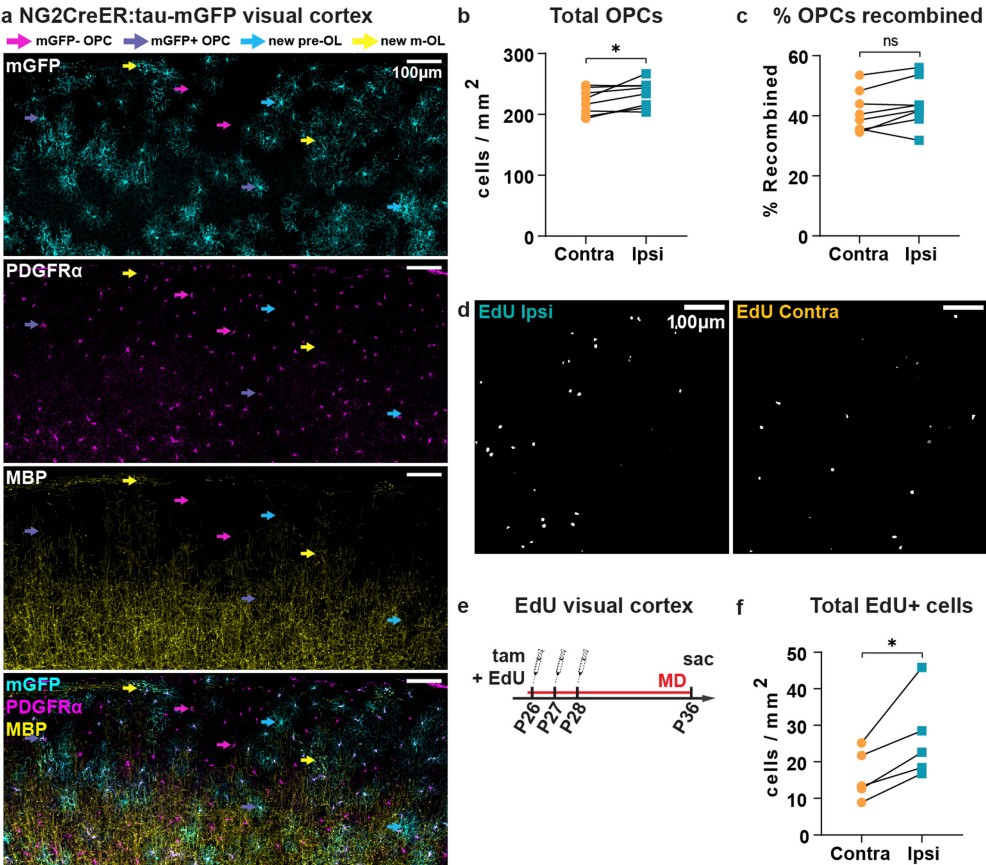

**Extended Data Fig. 1 | Sensory experience during adolescence modulates oligodendroglial dynamics.** (A) Example image from visual cortex. OPC = oligodendrocyte precursor cell, pre-OL = pre-myelinating oligodendrocyte, m-OL = mature oligodendrocyte. (B) Total PDGFRα + OPCs per hemisphere (p = 0.0373). Contra = hemisphere contralateral to the deprived eye, ipsi = hemisphere ipsilateral to the deprived eye. (C) Percentage of PDGFRα + OPCs that were mGFP+ (p = 0.066). (D) Example images of EdU+ cells in visual cortex. (E) Timeline of EdU injections. (F) Quantification of total EdU+ cells in both hemispheres of visual cortex (p = 0.022). Panels B-C: paired two-tailed t-test, n = 8 mice. Panel F: paired two-tailed t-test, n = 5 mice. *p < 0.05, ns = not significant. Additional statistical details in Table S1.

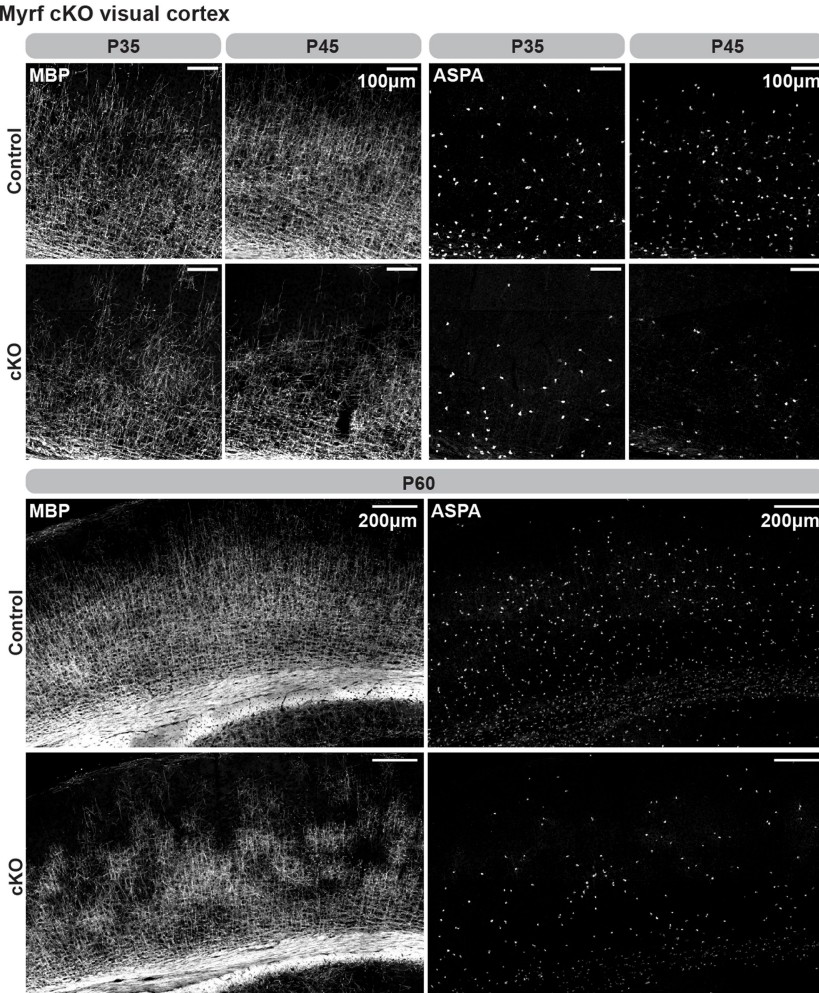

**Extended Data Fig. 2 | OPC-specific deletion of Myrf in adolescence impairs oligodendrogenesis and myelination in visual cortex.** Example images of myelin (MBP) and mature oligodendrocytes (ASPA) in visual cortex of control and Myrf cKO mice at P35, P45, and P60.

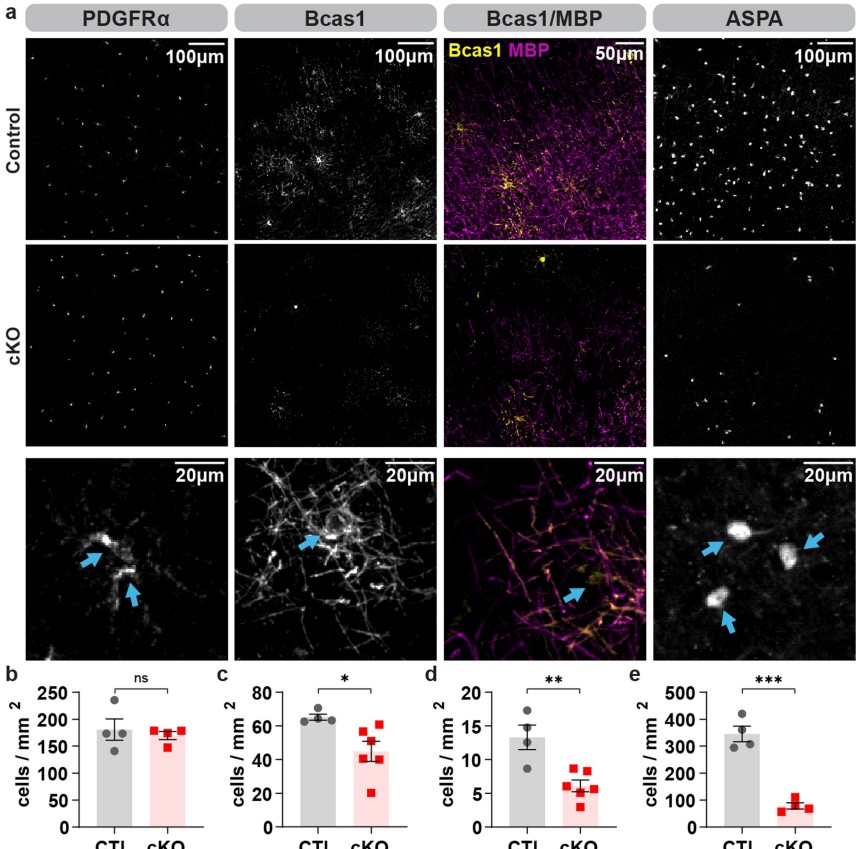

**Extended Data Fig. 3 | Stage-dependent decrease in oligodendrocyte density in adult visual cortex of Myrf cKO mice.** (A) Example images and (B) quantification of OPCS (PDGFRα; p = 0.6237), (C) newly formed oligodendrocytes (Bcas1; p = 0.0291), (D) newly formed oligodendrocytes that are actively myelinating (Bcas1/MBP; p = 0.004), and (E) mature oligodendrocytes (ASPA; p = 0.0001) of control and Myrf cKO mice at P60. Blue arrows denote cell bodies. Panels B-E: unpaired two-tailed t-tests, n = 4 CTL mice and 6 cKO mice. Data are presented as mean +/- SEM. *p < 0.05, **p < 0.01, ***p < 0.001, ns = not significant. Additional statistical details in Table S1.

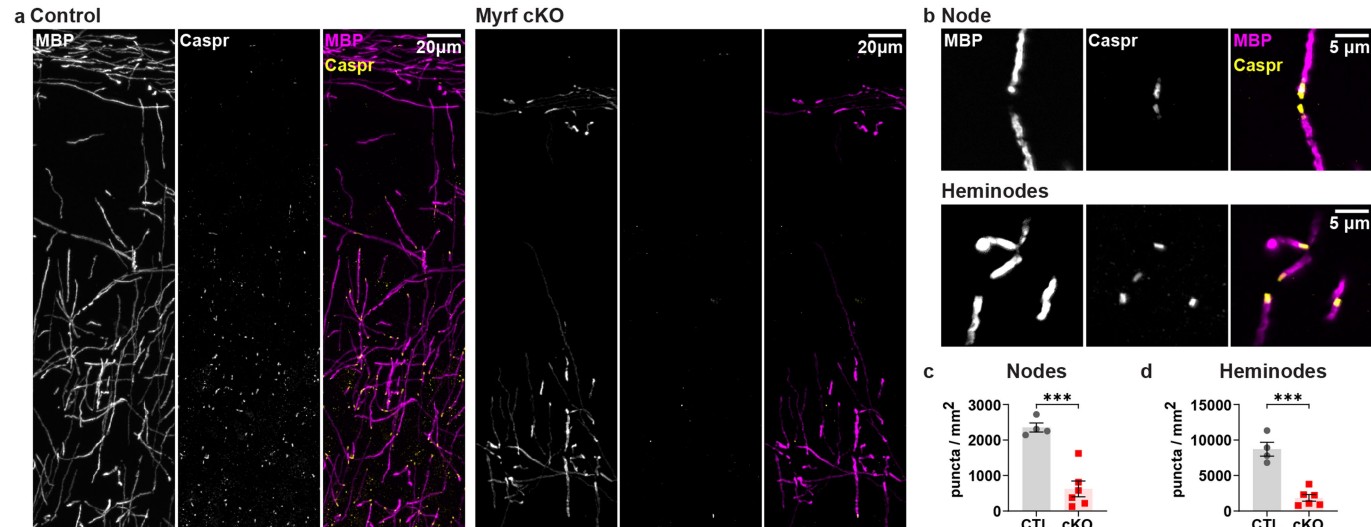

**Extended Data Fig. 4 | Analysis of Caspr+ paranodes in adult visual cortex of control and Myrf cKO mice.** (A) Example images of myelin sheaths (MBP) and paranodes (Caspr) in adult visual cortex. (B) Example images of pairs of Caspr+ puncta categorized as a node (top row) or as heminodes (bottom row). (C) Quantification of number of nodes (p = 0.0004) and (D) heminodes (p = 0.0001) by genotype. Panels C-D: unpaired two-tailed t-tests, n = 4 CTL mice and 6 cKO mice. Data are presented as mean +/- SEM. ***p < 0.001, ns = not significant. Additional statistical details in Table S1.

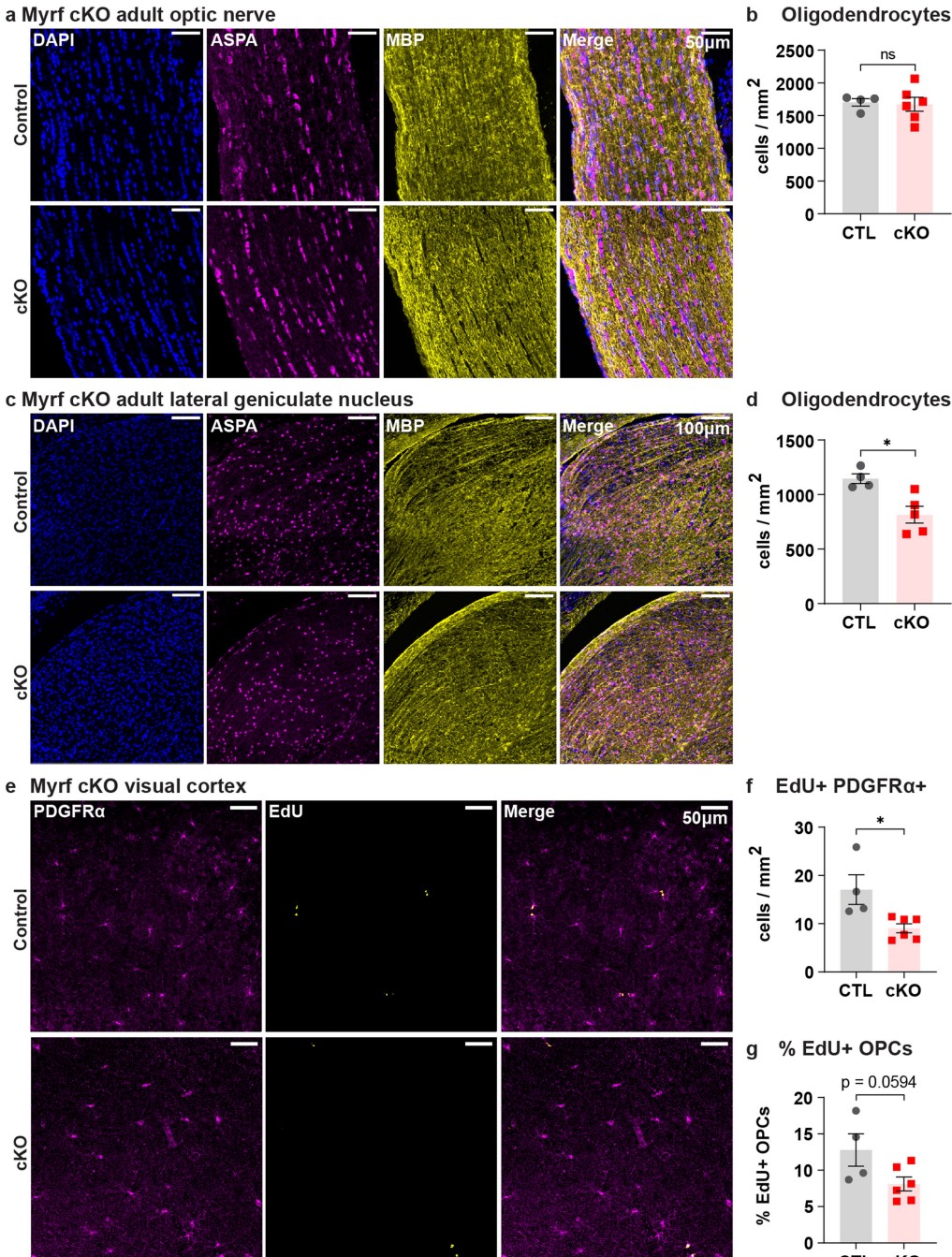

**a Myrf cKO adult optic nerve**

**b Oligodendrocytes**

**c Myrf cKO adult lateral geniculate nucleus**

**d Oligodendrocytes**

**e Myrf cKO visual cortex**

**f EdU+ PDGFRα+**

**g % EdU+ OPCs**

**Extended Data Fig. 5 | Myelination in optic nerve and lateral geniculate nucleus, and visual cortex OPC proliferation, of adult control and Myrf cKO mice.** (A) Example images of myelin (MBP) and oligodendrocytes (ASPA) in optic nerve. (B) Quantification of ASPA+ oligodendrocytes in optic nerve (p = 0.8477). (C) Example images of myelin and oligodendrocytes in lateral geniculate nucleus. (D) Quantification of ASPA+ oligodendrocytes in lateral geniculate nucleus (p = 0.0105). (E) Example images of OPCs (PDGFRα) and EdU labeling in visual cortex. (F) Quantification of PDGFRα + EdU+ OPCs in visual cortex (p = 0.0176). (G) Quantification of percentage of PDGFRα + OPCs that were EdU+ in visual cortex. Panels B, F-G: unpaired two-tailed t-tests, n = 4 CTL mice and 6 cKO mice. Panel D: unpaired two-tailed t-test, n = 4 CTL mice and 5 cKO mice. Data are presented as mean +/- SEM. *p < 0.05. Additional statistical details in Table S1. Data are presented as mean +/- SEM. *p < 0.05, ns = not significant. Additional statistical details in Table S1.

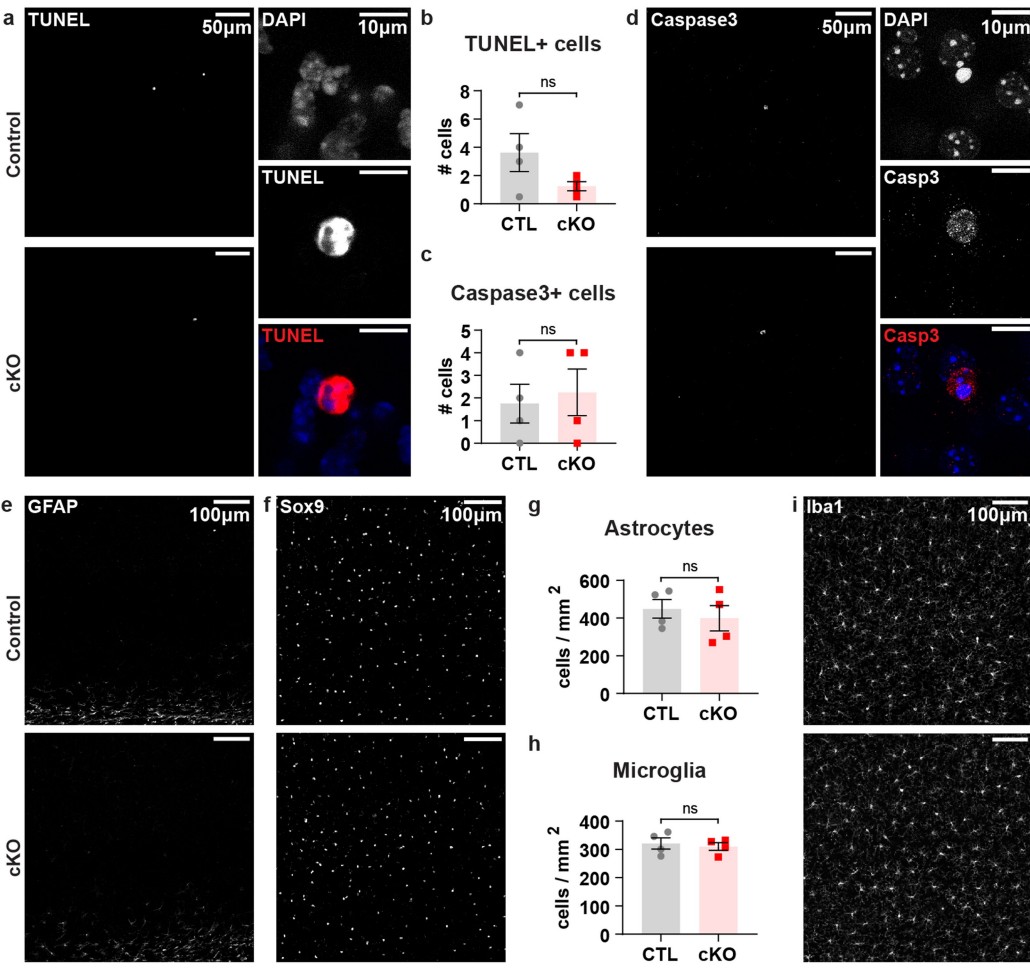

**Extended Data Fig. 6 | Cell death, astrocyte density, and microglia density in adult control and Myrf cKO mice.** (A) Example images and (B) quantification of TUNEL immunostaining in visual cortex (p = 0.1748). (C) Quantification (p = 0.7216) and (D) example images of Caspase+ cells in visual cortex. (E) Example images of GFAP immunostaining in visual cortex. (F) Example images and (G) quantification of Sox9+ astrocytes (p = 0.5736) in visual cortex. (H) Quantification (p = 0.6627) and (I) example images of Iba1+ microglia in visual cortex. Panel B: Welch's t-test, n = 4 CTL mice and 4 cKO mice. Panels C, G-H: unpaired two-tailed t-test, n = 4 CTL mice and 4 cKO mice. Data are presented as mean +/- SEM. ns = not significant. Additional statistical details in Table S1.

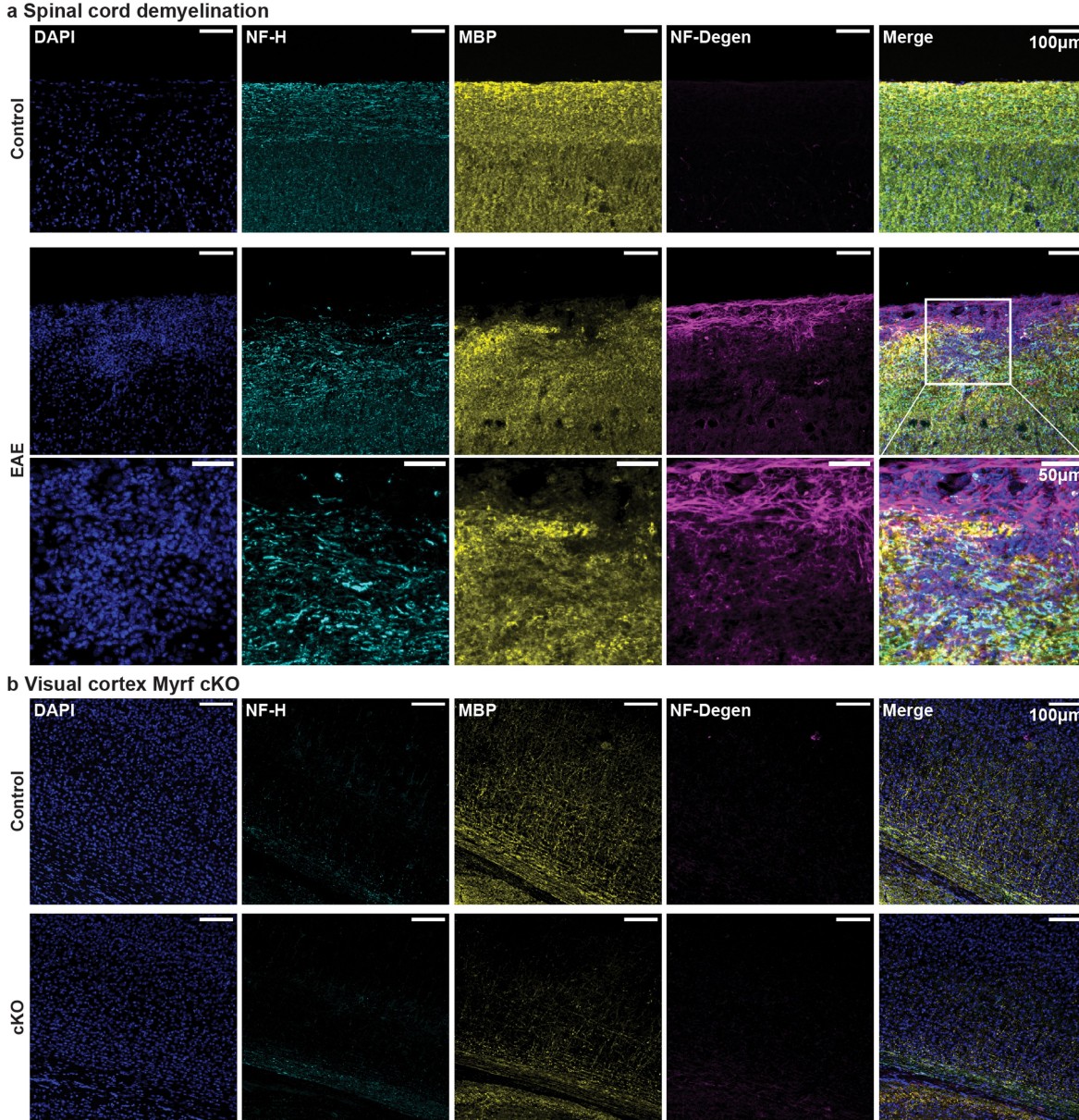

**a Spinal cord demyelination**

| DAPI | NF-H | MBP | NF-Degen | Merge |

**b Visual cortex Myrf cKO**

| DAPI | NF-H | MBP | NF-Degen | Merge |

**Extended Data Fig. 7 | Assessing neurodegeneration in demyelination and in Myrf cKO mice.** (A) Immunostaining for neurofilament H (NF-H), myelin basic protein (MBP), and neurofilament light chain DegenoTag (NF-Degen) in spinal cords of control mice and mice that underwent experimental autoimmune encephalitis (EAE). Disordered NF-H and prominent NF-Degen signal can be detected in the spinal cord of EAE mice, most notably in regions of demyelination. (B) Immunostaining for NF-H, MBP, and NF-Degen in visual cortex of control and Myrf cKO mice.

**a  Retinotopic organization in primary visual cortex**

**b  Contra eye response**

**c  Ipsi eye response**

**Extended Data Fig. 8 | Retinotopic organization and amplitude of visual cortex responses to visual stimulation in adult control and Myrf cKO mice.**
(A) Example intrinsic signal images of retinotopy in primary visual cortex.
(B, C) Amplitude of intrinsic signal imaging responses to stimulation of the contralateral deprived eye (contra; p < 0.0001) or ipsilateral non-deprived eye (ipsi; p < 0.0001) in binocular visual cortex of control (CTL) and cKO mice. Panels B-C: two-way ANOVA followed by Sidak's multiple comparisons test, n = 8 CTL mice and 9 cKO mice. Data are presented as mean +/- SEM. **p < 0.01, ***p < 0.001, ****p < 0.0001. Additional statistical details in Table S1.

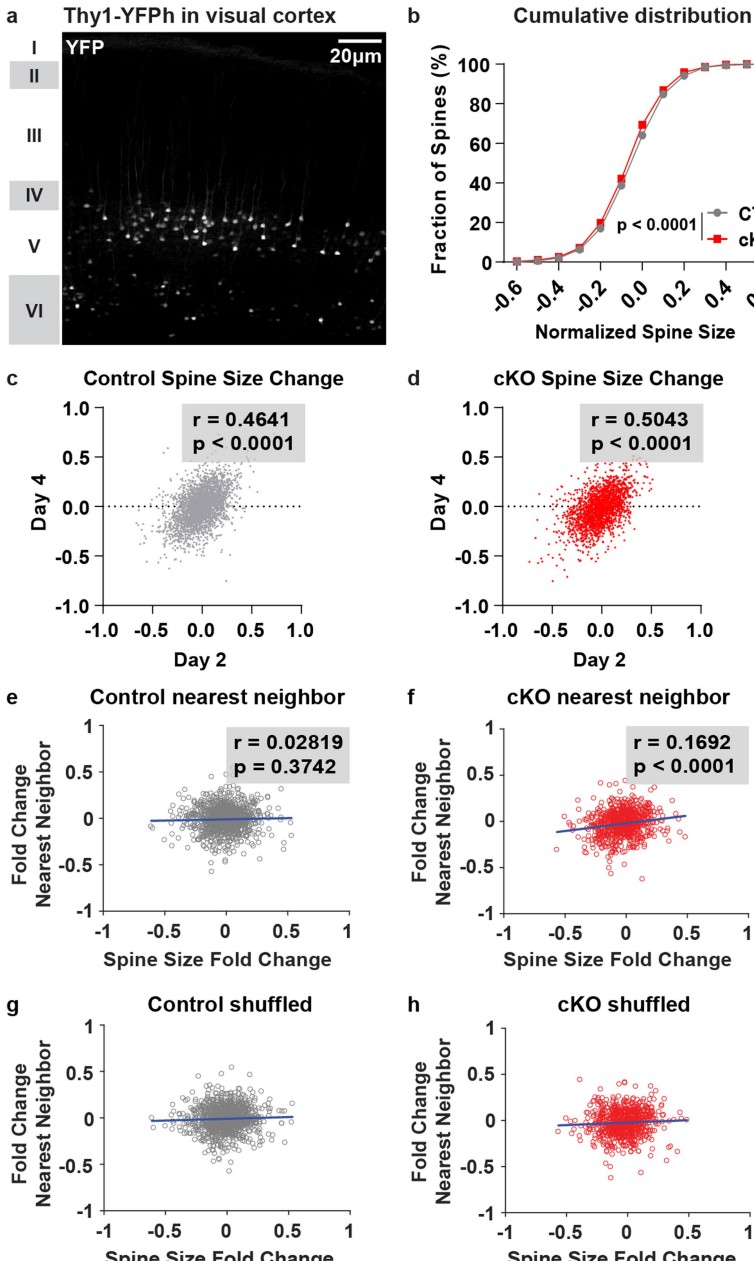

**Extended Data Fig. 9 | Spine size changes following monocular deprivation in adult control and Myrf cKO mice.** (A) Low magnification image of Thy1YFPh expression in visual cortex. (B) Cumulative distribution plot of spine size changes in control (CTL) and Myrf cKO (cKO) mice after four days of monocular deprivation. (C, D) Correlation of spine size changes after two days of monocular deprivation with spine size changes after four days of monocular deprivation in control and cKO mice. (E, F) Correlation of average size change following monocular deprivation for a given spine and size change of its nearest neighbor in control and cKO mice. (G, H) Example correlation of nearest neighbor spine changes in one set of shuffled spine pairings for control and cKO mice. Panel B: Kolmogorov-Smirnov test, n = 3484 spines from 10 CTL mice and 2438 spines from 10 cKO mice. Panels C-F: Pearson r correlation, n = 10 CTL and 10 cKO mice, exact numbers of spine numbers and spine pairs can be found along with additional statistical details in Table S1.

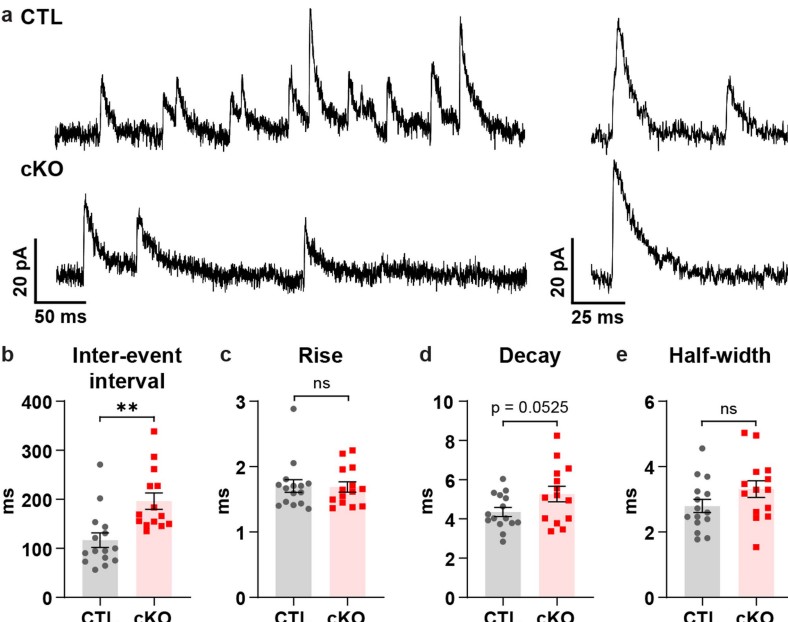

**Extended Data Fig. 10 | Inhibitory synaptic transmission in adult control and Myrf cKO mice.** (A) Example voltage clamp traces recorded at 0 mV from control (CTL) and Myrf cKO (cKO) mice, in the presence of TTX, APV, and NBQX. (B) Quantification of mIPSC inter-event interval (p = 0.0014), (C) rise time (p = 0.9116), (D) decay time (p = 0.0525), and (E) half-width (p = 0.1218). Panels B-E: unpaired two-tailed t-test, n = 15 cells from 5 CTL mice and 14 cells from 5 cKO mice. Data are presented as mean +/− SEM. **p < 0.01, ns = not significant. Additional statistical details in Table S1.

# Reporting Summary

## Statistics

For all statistical analyses, confirm that the following items are present in the figure legend, table legend, main text, or Methods section.

| n/a | Confirmed | |
|---|---|---|
| ☐ | ☒ | The exact sample size (*n*) for each experimental group/condition, given as a discrete number and unit of measurement |
| ☐ | ☒ | A statement on whether measurements were taken from distinct samples or whether the same sample was measured repeatedly |
| ☐ | ☒ | The statistical test(s) used AND whether they are one- or two-sided *Only common tests should be described solely by name; describe more complex techniques in the Methods section.* |
| ☐ | ☒ | A description of all covariates tested |
| ☐ | ☒ | A description of any assumptions or corrections, such as tests of normality and adjustment for multiple comparisons |
| ☐ | ☒ | A full description of the statistical parameters including central tendency (e.g. means) or other basic estimates (e.g. regression coefficient) AND variation (e.g. standard deviation) or associated estimates of uncertainty (e.g. confidence intervals) |
| ☐ | ☒ | For null hypothesis testing, the test statistic (e.g. *F*, *t*, *r*) with confidence intervals, effect sizes, degrees of freedom and *P* value noted *Give P values as exact values whenever suitable.* |
| ☐ | ☒ | For Bayesian analysis, information on the choice of priors and Markov chain Monte Carlo settings |
| ☒ | ☐ | For hierarchical and complex designs, identification of the appropriate level for tests and full reporting of outcomes |
| ☐ | ☒ | Estimates of effect sizes (e.g. Cohen's *d*, Pearson's *r*), indicating how they were calculated |

*Our web collection on statistics for biologists contains articles on many of the points above.*

## Software and code

Policy information about availability of computer code

| | |
|---|---|
| Data collection | In situ images were collected via proprietary Zeiss Zen 2 (blue edition, version 2.0.0.0) software. Intrinsic signal images were acquired using custom Linux software. In vivo images were acquired using ThorImage LS. Slice recordings were obtained with a Multiclamp 700B amplifier from Molecular Devices using WinWCP software. |
| Data analysis | Fluorescent images were analyzed using FIJI and MapManager software (https://mapmanager.net) written in Igor Pro (WaveMetrics) as previously described in Roth et al., Neuron, 2020. Intrinsic signal imaging data were analyzed using custom Matlab code as described in Cang et al., Visual Neuroscience, 2005. Slice recordings were analyzed using MiniAnalysis from Synaptosoft. Statistical analysis was performed using Matlab v2016b and Graphpad Prism v9. |

For manuscripts utilizing custom algorithms or software that are central to the research but not yet described in published literature, software must be made available to editors and reviewers. We strongly encourage code deposition in a community repository (e.g. GitHub). See the Nature Portfolio guidelines for submitting code & software for further information.

## Data

Policy information about availability of data

All manuscripts must include a data availability statement. This statement should provide the following information, where applicable:

- Accession codes, unique identifiers, or web links for publicly available datasets
- A description of any restrictions on data availability
- For clinical datasets or third party data, please ensure that the statement adheres to our policy

> No databases were generated during the course of this study. Raw data for any main or supplemental figure can be supplied upon request.

## Research involving human participants, their data, or biological material

Policy information about studies with human participants or human data. See also policy information about sex, gender (identity/presentation), and sexual orientation and race, ethnicity and racism.

| | |
|---|---|
| Reporting on sex and gender | N/A |
| Reporting on race, ethnicity, or other socially relevant groupings | N/A |
| Population characteristics | N/A |
| Recruitment | N/A |
| Ethics oversight | N/A |

Note that full information on the approval of the study protocol must also be provided in the manuscript.

# Field-specific reporting

Please select the one below that is the best fit for your research. If you are not sure, read the appropriate sections before making your selection.

☒ Life sciences      ☐ Behavioural & social sciences      ☐ Ecological, evolutionary & environmental sciences

For a reference copy of the document with all sections, see nature.com/documents/nr-reporting-summary-flat.pdf

# Life sciences study design

All studies must disclose on these points even when the disclosure is negative.

| | |
|---|---|
| Sample size | We used published standards from comparable studies to determine sample size for all experiments (e.g. for histology experiments, see Mei et al., JNeurosci, 2016; for intrinsic signal imaging, see Kaneko et al., eLife, 2014; for in vivo imaging of dendritic spines, see Qiao et al., Cell Reports, 2022; for slice electrophysiology, see Chokshi et al., Neuron, 2019). |
| Data exclusions | No data were excluded from analyses. |
| Replication | Reported Ns for each experiment reflect total number of biological replicates. Each experiment contains a minimum of two independent experiments. Results were consistent across separate experiments and no data were excluded from analyses. |
| Randomization | Mice were randomly assigned to cages upon weaning. All mice underwent the same experimental procedures. |
| Blinding | Experimenters were blinded to genotypes throughout data acquisition and analysis. |

# Reporting for specific materials, systems and methods

We require information from authors about some types of materials, experimental systems and methods used in many studies. Here, indicate whether each material, system or method listed is relevant to your study. If you are not sure if a list item applies to your research, read the appropriate section before selecting a response.

## Materials & experimental systems

| n/a | Involved in the study |
|-----|----------------------|
| ☐ | ☒ Antibodies |
| ☒ | ☐ Eukaryotic cell lines |
| ☒ | ☐ Palaeontology and archaeology |
| ☐ | ☒ Animals and other organisms |
| ☒ | ☐ Clinical data |
| ☒ | ☐ Dual use research of concern |
| ☒ | ☐ Plants |

## Methods

| n/a | Involved in the study |
|-----|----------------------|
| ☒ | ☐ ChIP-seq |
| ☒ | ☐ Flow cytometry |
| ☒ | ☐ MRI-based neuroimaging |

# Antibodies

| Antibodies used | Antibody Source Identifier Concentration |
|-----------------|------------------------------------------|
| | Rabbit anti-ASPA (clone N1C3-2) GeneTex Cat# GTX113389; RRID AB_2036283 1:1000 |
| | Chicken anti-GFP Rockland Cat# 600-901-215; RRID AB_1537403 1:1000 |
| | Rat anti-MBP Millipore (clone 12) Cat# MAB386; RRID AB_94975 1:200 |
| | Rabbit anti-PDGFRα W.B. Stallcup N/A 1:200 |
| | Rabbit anti-cleaved Caspase3 (Asp175) Cell Signaling Cat# 9661S; RRID AB_2341188 1:200 |
| | Mouse anti-GFAP (clone GA5) Millipore Cat# MAB360; RRID AB_11212597 1:1000 |
| | Human anti-Sox9 R&D Systems Cat# AF3075; RRID AB_2194160 1:2000 |
| | Rabbit anti-Iba1 Wako Cat# 019-19741; RRID AB_839504 1:1000 |
| | Mouse anti-NF-L Degenotag Encor Cat# MCA-1D44; RRID AB_2923483 1:1000 |
| | Rabbit anti-NF-H Abcam Cat# ab8135; RRID AB_306298 1:1000 |
| | Mouse anti-PV (clone 235) Swant Cat# 235; RRID AB_10000343 1:1000 |
| | Biotinylated WFA Vector Labs Cat# B-1355; RRID AB_2336874 1:400 |
| | Rabbit anti-Caspr Peles lab N/A 1:600 |
| | Mouse anti-Bcas1 (5) Santa Cruz Cat# SC-136342; RRID AB_10839529 1:300 |
| | Goat anti-rabbit AlexaFluor 488 Thermo Fisher Scientific Cat# A-11034; RRID:AB_2576217 1:1000 |
| | Goat anti-rabbit AlexaFluor 594 Thermo Fisher Scientific Cat# A-11012; RRID:AB_2534079 1:1000 |
| | Goat anti-rabbit AlexaFluor 647 Thermo Fisher Scientific Cat # A-21245; RRID:AB_2535813 1:1000 |
| | Goat anti-chicken AlexaFluor 488 Thermo Fisher Scientific Cat # A-11039; RRID:AB_142924 1:1000 |
| | Goat anti-rat AlexaFluor 647 Thermo Fisher Scientific Cat# A-21247; RRID:AB_141778 1:1000 |
| | Goat anti-rat AlexaFluor 488 Jackson ImmunoResearch Cat# 112-545-167; RRID:AB_2338362 1:1000 |
| | Goat anti-mouse AlexaFluor 488 Jackson ImmunoResearch Cat# 115-545-166; RRID:AB_2338852 1:1000 |
| | Goat anti-mouse AlexaFluor 647 Thermo Fisher Scientific Cat# A-21236; RRID:AB_2535805 1:1000 |
| | Goat anti-human AlexaFluor 594 Thermo Fisher Scientific Cat# A-11014; RRID:AB_2534081 1:1000 |
| | Alexa Fluor® 594 Streptavidin Jackson ImmunoResearch Cat# 016-580-084; RRID:AB_2337250 1:1000 |
| Validation | Primary antibodies used in this study were previously validated by manufacturers for specificity and/or were previously published by our lab and others. Rabbit anti-ASPA antibody validation: positive control - ASPA-transfected 293T (performed by manufacturer); also see Xin et al., Cell Reports 2019 and Larson et al., eLife 2018 for additional applications. Chicken anti-GFP antibody validation: negative control - absence of signal in GFP-negative tissue (performed in Chan lab). Rat anti-MBP antibody: see Haines et al. Nature Neuroscience 2015 and Lodato et al., Nature Neuroscience 2014 for additional applications. Rabbit anti-PDGFRα validation: negative control - knockout tissue (performed in Chan lab); see Mayoral et al., Cell Reports 2018 for additional applications. Rabbit anti-cleaved Caspase3 antibody: used in over 9000 citations; see Bhadury et al., Oncogenesis 2013 and Sun et al., Nature Communications 2023 for additional applications. Mouse anti-GFAP antibody: manufacturer's statement "Anti-Glial Fibrillary Acidic Protein Antibody, clone GA5 detects level of Glial Fibrillary Acidic Protein & has been published & validated for use in IC, IH, IH(P) & WB with more than 65 product citations". Human anti-Sox9 antibody: see O'Shea et al., Nature Communications 2022 and Huang et al., Neuron 2020 for additional applications. Rabbit anti-Iba1 antibody: has been used in over 4000 citations; see Monai et al., Nature Communications 2016 and Noguchi et al., Communications Biology 2024 for additional applications. Mouse anti-NF-L Degenotag antibody: see Shaw et al., Brain Communications 2023 for validation by manufacturer. Rabbit anti-NF-H antibody: see Kodati et al, Biomedicines 2022 and Kulesskaya et al., Frontiers in Cell and Developmental Biology 2022 for additional applications. Mouse anti-PV antibody: see Hoseini et al., eLife 2021 and Berdugo Vega et al., Nature Communications 2020 for additional applications. Biotinylated WFA antibody: see Okur et al., Nature 2024 and Lupori et al., Cell Reports 2023 for additional applications. Rabbit anti-Caspr antibody: previously validated in knockout tissue (performed in Peles and Chan labs); see Chang et al., eLife 2021 and Eshed-Eisenbach et al., Neuron 2020 for additional applications. Mouse anti-Bcas1 antibody: see Cunha et al., Journal of Experimental Medicine 2020 and Maas et al., Nature Communications 2020 for additional applications. |

# Animals and other research organisms

Policy information about studies involving animals; ARRIVE guidelines recommended for reporting animal research, and Sex and Gender in Research

| Laboratory animals | NG2CreER:tau- mGFP mice (Zhu et al. 2011, Jax # 008538; Hippenmeyer et al. 2005, Jax # 021162), aged P26-35<br>PDGFRαCreER:MyrfFl/Fl mice (Kang et al. 2010, Jax # 018280; Emery et al. 2009, Jax # 010607), aged P21-90<br>Thy1-YFP-H mice (Feng et al., 2000, Jax # 003782), aged P60-90 |
|--------------------|-------------------------------------------------------------------------------------------------------------|
| Wild animals | No wild animals were used in this study. |

| Reporting on sex | Both male and female mice were used in this study. Due to the challenge of obtaining sufficient numbers of mice with the appropriate genetic crosses, groups were not balanced by sex. |
| --- | --- |
| Field-collected samples | No field-collected samples were used in this study. |
| Ethics oversight | All mice were handled in accordance with, and all procedures approved by, the Institutional Animal Care and Use Committee of the University of California San Francisco. |

Note that full information on the approval of the study protocol must also be provided in the manuscript.

## Plants

| Seed stocks | N/A |
| --- | --- |
| Novel plant genotypes | N/A |
| Authentication | N/A |

