## [Peer Review File · Nature]

Manuscript Title: Oligodendrocytes and myelin limit neuronal plasticity in visual cortex

Reviewer Comments & Author Rebuttals

Reviewer Reports on the Initial Version:

Referees' comments:

Referee #1 (Remarks to the Author):

The manuscript from Xin and colleagues examines the role that oligodendrocytes and myelin play in sensory neuron plasticity. The study tests the hypothesis that myelination, which occurs relatively late in development, stabilises neuronal circuits and dampens neuronal plasticity. To examine this, the investigators focus on the visual cortex and use ocular deprivation in combination with a genetic approach to block myelination to examine neuronal plasticity. Sensory neurons in adult mice in which myelination was blocked during postnatal development display increased plasticity relative to controls. This is correlated with a change in the dendritic spine response to ocular deprivation. Overall, these are sophisticated studies that examine a critical area of developmental neuroscience related to the impact of myelinating glial cells on neuronal plasticity. The results support the premise that myelin suppresses neuronal plasticity and should be of considerable interest. There are areas, however, where the study could be improved, as highlighted below.

In figure 1 it would be helpful to add a cell body marker for mature oligodendrocytes (e.g., ASPA, CC1) so that these cells can be more easily discerned and quantified.

If the newly formed oligodendrocytes (Figure 1) are derived from OPCs that had not first divided, what cell type replaces the OPCs to maintain their homeostasis? Has this been observed before, and if so, it should be noted with a reference.

The absence of an OPC mitotic response when normal oligodendrocyte differentiation is blocked in the Myrf cKO mice is perplexing and goes against the concept that the loss of oligodendrocytes leads to their replacement by OPCs. Has this been observed previously, and if so, this should be discussed.

The authors should consider using a protein marker (e.g., BCAS1, Enpp6) to identify newly formed pre-myelinating oligodendrocytes (Figure 1), as opposed to the absence of OPC and mature oligodendrocyte markers.

In the Myrf cKO mice what happens to the newly differentiating oligodendrocytes? Emery et al. (PMID: 19596243) show that the mutant cells die by apoptosis during development, but the authors do not detect an increase in apoptotic cells in the mutants. Some effort to characterise their fate would be informative. The possibility that the fate of these cells (e.g., widespread necrosis) could have an indirect

impact on the results should be discussed.

Throughout, the analysis of myelin is somewhat lacking, despite being highlighted in the manuscript title. Some effort to quantify the impact of ocular deprivation on myelin content in WT and Myrf mutants would provide additional support for the premise of the manuscript.

The recently published manuscript (PMID: 37838794) describing a critical role for newly formed oligodendrocytes and myelination in non-motor learning and cognition is related and should be referenced.

Referee #2 (Remarks to the Author):

The paper by Xin et al. shows that oligodendrocyte maturation and myelination limit plasticity in adult visual cortex. Using conditional expression of GFP in oligodendrocyte precursor cells, the authors show that sensory deprivation in juvenile mice reduces oligodendrocyte maturation and myelination in the visual cortex. Next, they genetically blocked oligodendrocyte differentiation by conditionally knocking out the transcription factor Myrf, causing reduced myelination in the visual cortex. Adult plasticity following monocular deprivation (MD) was strongly altered in these mice: while controls showed the expected pattern of eye-specific changes, i.e. an increase in open eye strength, but no concomitant drop in closed eye responses, the KO mice showed a clearly juvenile pattern, with a strong drop in closed eye responses. Finally, they used chronic structural imaging of dendritic spines, revealing a clear difference between genotypes during MD. These differences were less pronounced when analyzing spine turnover, but were prominent when the authors assessed spine size, a proxy for synapse strength. In particular, mice with impaired myelination showed spatially coordinated spine size changes, with neighboring spines frequently changing in the same direction, i.e. jointly growing or shrinking.

The adult mammalian brain is less plastic than the juvenile one. Typically, plasticity is particularly high during so-called critical periods, whose timing differs between brain regions. A number of factors ending the critical period and limiting adult plasticity have been identified, but the picture is far from complete. Myelin and molecules associated with it have long been shown to play an important role in limiting axonal growth and plasticity in the adult mammalian brain. This paper shows that the proper maturation of the myelin-producing cells, oligodendrocytes, is also crucial for controlling plasticity in one of the most commonly used models for experience dependent plasticity, monocular deprivation. The paper provides substantial new insights by showing that the effect of myelin is tightly linked to specific aspects of structural plasticity in the visual cortex. The results are very clearly and concisely presented and altogether are very convincing.

While I am overall very positive about this study, there are several points which the authors should address in a revised version.

Major:

1. In contrast to control mice, adult KO mice showed a juvenile-like form of OD plasticity, with clear closed eye depression. However, a number of studies (e.g. Matthies et al., 2013; Löwel lab) have shown that also normal adult mice can show strong closed eye depression, when they are kept under specific conditions during MD, e.g. enriched environment, continuous visual stimulation... How were the mice in the present study kept? Group housing is mentioned, but it is not clear whether this was true also during MD. Was there any environmental enrichment?
2. The MD induced spine density and turnover changes shown in Figure 3 for control mice are substantially different from those reported by Hofer et al., 2009. The authors should comment on this discrepancy.
3. Is there any chance to correlate the magnitude of the OD shift with that of the spine size changes per mouse? Related, what was the reason for following spine changes over just four days, while the MD effects are clearly stronger after eight days?

Minor:

4. In the abstract, the authors write “visual cortex activity following monocular deprivation remained stable”. This is not what they show, rather, there was an increase in open eye responses.
5. Figure 1m: given these very low numbers, I am not sure whether one can claim that the number “was the same in both hemispheres”. At the very least, this should be phrased more carefully. How many cells were actually counted for this analysis?
6. It should be indicated on how many sections, or um, the maximum intensity projections in Figure 3 are based on.
7. It is not clear what the right square bracket in Figure 4a indicates.

Referee #3 (Remarks to the Author):

This study's aim is to determine if developmental myelination restricts neuronal plasticity. They do this by blocking oligodendrocyte maturation, thereby reducing myelination, in adolescent mice, and then subsequently assessing functional and structural experience-induced neuronal plasticity with monocular deprivation in the adult visual cortex. The authors demonstrate that blocking normal oligo developmental maturation results in adult visual cortical circuit's response to visual stimulation being significantly weakened. This adult weakened circuit exhibits pronounced decreases in response following visual deprivation that are not found in control adult animals. This is the basis for the authors conclusion that there is a key role for adolescent oligodendrogenesis and myelin formation in restricting functional experience-dependent neuronal plasticity in the visual cortex.

Major Concern:

The authors have shown a clear effect of their developmental manipulation of oligodendrocytes on both

the state of the cortical circuits and on their response to deprivation. But the conclusion that is drawn 'myelination restricts plasticity' implies that this is the only aspect of cortical organization that has been impacted by the manipulation, when it's clear that the functional synaptic organization of the circuit in animals that have undergone this manipulation is far from normal. This could be interesting if there was an effort to define how this manipulation was actually altering the circuit, but beyond 'weaker responses' and 'lower spine density', it's not clear exactly how the circuit has been altered, the mechanisms that underly this alteration, and how these alterations contribute to the effects of monocular deprivation that are seen. As it is presented, the only conclusion that can be derived is that altering oligodendrocyte maturation and myelination prevents the normal maturation of the cortex, and leaves the cortex in a state in which monocular deprivation exhibits an effect. This is far from strong, direct evidence that myelination restricts plasticity, which is the author's claim.

Other issues:

1. Previous studies have implicated inhibitory circuits as playing a significant role in regulating the period of visual cortex plasticity. While the authors show that the number of parvalbumin neurons, the candidate inhibitory cell type most expected to be involved, does not decrease, there is no direct test of the level of inhibition present in the altered circuit, and this remains one of many possible explanations for the presence of adult plasticity.

2. The measures of activity changes to visual stimulation were made with an indirect approach--intrinsic signal imaging—which measures changes in the oxy/deoxyhemoglobin ratio, i.e. a vascular change rather than a change in the activity of individual neurons. This seems odd given all the work of the last 15 years showing the greatly enhanced sensitivity and resolution of genetically encoded calcium sensors, which can be used at the episcala and at the cellular scale with 2-photon imaging to provide a much better measure of the neuronal activity patterns. Even stranger given that the authors used 2-P in vivo imaging to describe morphological changes.

3. The measurement of spine numbers here are from the apical dendrites of L5 pyramidal neurons. The authors need to address why L5 pyramidal neurons were chosen for observing plasticity changes in visual cortex.

4. The significance of the data displayed in the graphs of Figure 3d-h is not apparent. If the goal was to show differences in spine dynamics following deprivation between control animals and those with impaired oligodendrogenesis, the reader walks away with the exact opposite impression. Differences between control and knockout are present before and following monocular deprivation, so it is not clear why this analysis is of value in judging impact of monocular deprivation. Indeed, one change that appears to occur as a result of deprivation is an increase in the size of the spines in the control animals (which are not thought to exhibit plasticity) while the knockout animals (that are supposed to exhibit plasticity) appear to not increase in size.

4. Figure 2j- why is there an increase in the ipsi response in the control MD condition? This is not consistent with previous findings.

5. Figure 4 places changes in spine size in the context of dendritic topology, and it is really not clear why this analysis has been performed after the data in Figure 3 presents no clear picture of overall differences in spine morphology as a result of deprivation.

Author Rebuttals to Initial Comments:

We thank all the reviewers for their thoughtful and insightful comments. The comments were greatly encouraging and constructive. We think that this process has improved our manuscript and made our findings much more compelling. We have worked hard to perform numerous new experiments to address reviewer comments, adding several new figures and panels (Figure 5 and Extended Data Figures 3, 4, S7a-b, and 11), which we believe strengthen our conclusions and offer new insights. All changes in the manuscript text are highlighted.

Referee #1 (Remarks to the Author):

The manuscript from Xin and colleagues examines the role that oligodendrocytes and myelin play in sensory neuron plasticity. The study tests the hypothesis that myelination, which occurs relatively late in development, stabilises neuronal circuits and dampens neuronal plasticity. To examine this, the investigators focus on the visual cortex and use ocular deprivation in combination with a genetic approach to block myelination to examine neuronal plasticity. Sensory neurons in adult mice in which myelination was blocked during postnatal development display increased plasticity relative to controls. This is correlated with a change in the dendritic spine response to ocular deprivation. Overall, these are sophisticated studies that examine a critical area of developmental neuroscience related to the impact of myelinating glial cells on neuronal plasticity. The results support the premise that myelin suppresses neuronal plasticity and should be of considerable interest. There are areas, however, where the study could be improved, as highlighted below.

We thank the reviewer for their positive evaluation of our manuscript, and for their thoughtful suggestions for improving our study. Please find our specific responses below:

In figure 1 it would be helpful to add a cell body marker for mature oligodendrocytes (e.g., ASPA, CC1) so that these cells can be more easily discerned and quantified.

We agree with the reviewer that a cell body marker for mature oligodendrocytes would be a good addition to our existing analysis. Unfortunately, we were not able to successfully combine our genetic lineage tracing of tau-mGFP with ASPA immunostaining due to the technical requirement of antigen retrieval, which occluded the visualization of GFP by endogenous fluorescence and antibody amplification. We find that CC1 immunoreactivity is highly unreliable in visual cortex at this age, both in terms of specificity (i.e. many non-oligodendrocytes labeled) and penetrance (too few oligodendrocyte-appearing cells). However, we would like to point out that lineage tracing with tau-mGFP has been previously used to identify newly generated oligodendrocytes effectively (e.g. Young et al., 2012 and Wang et al., 2020^{1,2}). By modifying this approach with quantification of PDGFRa+/tau-mGFP+ cells (OPCs), PDGFRa-/MBP-/tau-mGFP+ cells (newly formed pre-myelinating oligodendrocytes) and MBP+/tau-mGFP+ cells (newly formed myelinating oligodendrocytes), we effectively capture the entire

lineage for oligodendroglial maturation. This approach clearly demonstrates that the significant change is not found in the pre-myelinating oligodendrocyte population but rather the newly generated myelinating oligodendrocytes that have integrated. In our opinion, this is a definite strength over ASPA or CC1 staining alone. We have added these additional references to contextualize our use of the tau-mGFP lineage tracing approach (Page 2, lines 59-61).

If the newly formed oligodendrocytes (Figure 1) are derived from OPCs that had not first divided, what cell type replaces the OPCs to maintain their homeostasis? Has this been observed before, and if so, it should be noted with a reference.

Indeed, our results are consistent with previous *in vivo* imaging studies that track the generation of oligodendrocytes in cortex (Hughes et al. 2013, 2018^{3,4}). Newly formed oligodendrocytes are most likely to arise from previously quiescent OPCs, and neighboring OPCs divide to replace the new differentiated cell. We have now added additional text in the manuscript to highlight these references (Page 2-3, lines 89-90).

The absence of an OPC mitotic response when normal oligodendrocyte differentiation is blocked in the *Myrf* cKO mice is perplexing and goes against the concept that the loss of oligodendrocytes leads to their replacement by OPCs. Has this been observed previously, and if so, this should be discussed.

Indeed, this pattern is consistent both with what we observed in our own data following sensory deprivation (Figure 1 - fewer mature oligodendrocytes in the contralateral cortex is accompanied by less OPC proliferation), as well as data from Hughes et al.³ demonstrating that in the healthy adult brain, oligodendrocyte differentiation is tightly linked to proliferation. Thus, outside the context of demyelination (which seems to induce an injury related OPC proliferation response), when there is lower differentiation with no loss of an existing oligodendrocyte, there is lower drive for the OPC pool to proliferate. We have added additional discussion on this topic in the text (Page 6, lines 245-248).

The authors should consider using a protein marker (e.g., BCAS1, Enpp6) to identify newly formed pre-myelinating oligodendrocytes (Figure 1), as opposed to the absence of OPC and mature oligodendrocyte markers.

We thank the reviewer for this suggestion; however, we are certain the reviewer appreciates the fact that the expression of any single marker along the oligodendroglial lineage is not necessarily definitive or exclusive from other premature or mature markers. We were unsure about the expression of BCAS1 and Enpp6 along the lineage and whether these populations of cells were committed precursors, newly differentiated oligodendrocytes, or myelinating oligodendrocytes. It is important to note that in our lineage tracing experiment, the significant change after monocular deprivation was not found in the pre-myelinating oligodendrocyte pool but rather the newly generated myelinating oligodendrocytes--actively expressing MBP. We have now provided additional data with BCAS1 staining in the *Myrf* cKO

mice to directly compare expression with MBP and ASPA (Extended Data Figure 3). While the BCAS1+/MBP+ cells (newly generated myelinating oligodendrocytes) are clearly reduced by almost 3-fold in the Myrf cKO mice, the BCAS1+/MBP- cells are only slightly reduced—and neither reduction is as dramatic as the change observed with the ASPA+ cells (Figure 2b, e; Extended Data Figures 2, 3). We conclude that BCAS1 is indeed expressed in newly formed pre-myelinating oligodendrocytes as well as a small population of newly formed myelinating oligodendrocytes. Total BCAS1 quantification would probably not be significantly altered after monocular deprivation. Based on the past and current literature, Enpp6 would be expressed even earlier than BCAS1 and would not necessarily help in our characterization of the changes observed after monocular deprivation.

In the Myrf cKO mice what happens to the newly differentiating oligodendrocytes? Emery et al. (PMID: 19596243) show that the mutant cells die by apoptosis during development, but the authors do not detect an increase in apoptotic cells in the mutants. Some effort to characterise their fate would be informative. The possibility that the fate of these cells (e.g., widespread necrosis) could have an indirect impact on the results should be discussed.

We thank the reviewer for bringing up this important point and have now included additional characterization of different stages of oligodendrocyte maturation in control and Myrf cKO mice (Extended Data Figure 3), as well as analysis of mature myelin segments as assessed by quantification of Caspr+ paranodes (Extended Data Figure 4). Furthermore, we performed TUNEL staining as an additional measure to detect cell death and found no clear differences between control and cKO mice (Extended Data Figure 7a-b). Based on the lack of increase in TUNEL+ or Caspase+ cells and no apparent signs of gliosis (Extended Data Figure 7) or axon degeneration (Extended Data Figure 8) (and no sign of widespread cell death by electron microscopy, see example images below), we do not believe there is widespread necrosis occurring in Myrf cKO mice. With that said, we agree with the reviewer about a potential increase in apoptosis over the course of adolescence due to a lack of mature oligodendrocyte integration and have included additional relevant discussion (Page 6, lines 249-253).

Figure 1 - Electron microscopy images from adult visual cortex of control and Myrf cKO mice.

Throughout, the analysis of myelin is somewhat lacking, despite being highlighted in the manuscript title. Some effort to quantify the impact of ocular deprivation on myelin content in WT and Myrf mutants would provide additional support for the premise of the manuscript.

We have now included additional characterization of different stages of oligodendrocyte maturation in control and Myrf cKO mice (Extended Data Figure 3). Based on our prior experience, simple immunostaining for myelin proteins is not a quantitative method for assessing myelination as the signal intensity reaches a saturating level and in fact, seems to decrease with increased myelination. As an alternative, we performed analysis of Caspr⁺ paranodes as a more quantitative proxy for the number of mature myelin segments (Extended Data Figure 4). We find that both mature nodes and heminodes are greatly diminished in the Myrf cKO mice—suggesting that myelin content is significantly reduced. We would like to thank the reviewer for this comment as our new data further strengthens our conclusions.

The recently published manuscript (PMID: 37838794) describing a critical role for newly formed oligodendrocytes and myelination in non-motor learning and cognition is related and should be referenced.

We thank the reviewer for pointing out this relevant reference and have added it to our discussion section (Page 6, lines 255-256).

Referee #2 (Remarks to the Author):

The paper by Xin et al. shows that oligodendrocyte maturation and myelination limit plasticity in adult visual cortex. Using conditional expression of GFP in oligodendrocyte precursor cells, the authors show that sensory deprivation in juvenile mice reduces oligodendrocyte maturation and myelination in the visual cortex. Next, they genetically blocked oligodendrocyte differentiation by conditionally knocking out the transcription factor *Myrf*, causing reduced myelination in the visual cortex. Adult plasticity following monocular deprivation (MD) was strongly altered in these mice: while controls showed the expected pattern of eye-specific changes, i.e. an increase in open eye strength, but no concomitant drop in closed eye responses, the KO mice showed a clearly juvenile pattern, with a strong drop in closed eye responses. Finally, they used chronic structural imaging of dendritic spines, revealing a clear difference between genotypes during MD. These differences were less pronounced when analyzing spine turnover, but were prominent when the authors assessed spine size, a proxy for synapse strength. In particular, mice with impaired myelination showed spatially coordinated spine size changes, with neighboring spines frequently changing in the same direction, i.e. jointly growing or shrinking.

The adult mammalian brain is less plastic than the juvenile one. Typically, plasticity is particularly high during so-called critical periods, whose timing differs between brain regions. A number of factors ending the critical period and limiting adult plasticity have been identified, but the picture is far from complete. Myelin and molecules associated with it have long been shown to play an important role in limiting axonal growth and plasticity in the adult mammalian brain. This paper shows that the proper maturation of the myelin-producing cells, oligodendrocytes, is also crucial for controlling plasticity in one of the most commonly used models for experience dependent plasticity, monocular deprivation. The paper provides substantial new insights by showing that the effect of myelin is tightly linked to specific aspects of structural plasticity in the visual cortex. The results are very clearly and concisely presented and altogether are very convincing.

While I am overall very positive about this study, there are several points which the authors should address in a revised version.

We thank the reviewer for their positive evaluation of our manuscript and have included specific responses below:

Major:

1. In contrast to control mice, adult KO mice showed a juvenile-like form of OD plasticity, with clear closed eye depression. However, a number of studies (e.g. Matthies et al., 2013; Löwel lab) have shown that also normal adult mice can show strong closed eye depression, when they are kept under specific conditions during MD, e.g. enriched environment, continuous visual stimulation... How were the mice in

the present study kept? Group housing is mentioned, but it is not clear whether this was true also during MD. Was there any environmental enrichment?

We have now added additional details about mouse husbandry and housing to the methods section of our paper (Page 6, lines 268-270). Mice were not housed in environmental enrichment beyond the normal IACUC mandated housing conditions. Since both control mice and Myrf cKO mice were housed in identical conditions, and control mice, like those studied previously (Sato and Stryker, 2008⁵), do not show ectopic functional plasticity in adulthood, we do not believe environmental conditions can explain the enhanced plasticity we observed in mice lacking adolescent oligodendrogenesis.

2. The MD induced spine density and turnover changes shown in Figure 3 for control mice are substantially different from those reported by Hofer et al., 2009. The authors should comment on this discrepancy.

We thank the reviewer for pointing out this discrepancy. We suspect the difference in spine dynamics we observed vs. what was reported by Hofer et al. may be driven by the difference in neuronal populations imaged – Hofer et al. used the Thy1-GFP-M line, whereas we imaged the Thy1-YFP-H line, which expresses YFP in a slightly different subset of pyramidal neurons (see example images of reporter expression in visual cortex of THY1-GFP-M vs THY1-YFP-H lines maintained in our laboratory below). Furthermore, reporter expression is known to drift over generations in Thy1 lines, which may further contribute variability in the population of neurons imaged. Hofer et al. reported a layer-specific effect of MD on spine dynamics in the visual cortex, so a different makeup of apical dendrites imaged could lead to a shift in the rate of spine increases and decreases observed with MD. We have now included a low magnification image of Thy1-YFP-H reporter expression in visual cortex in Extended Data Figure 10.

Figure 2 - Differential expression of fluorescent reporter in Thy1-GFP-M vs Thy1-YFP-H lines in visual cortex.

3. Is there any chance to correlate the magnitude of the OD shift with that of the spine size changes per mouse? Related, what was the reason for following spine changes over just four days, while the MD effects are clearly stronger after eight days?

We thank the reviewer for this excellent suggestion. Unfortunately, the two-photon imaging and intrinsic signal imaging were performed in separate animals, so we cannot perform a correlation between the two measures in our existing dataset. We do intend to follow up on this work with additional imaging on longer timescales and will take the reviewer's suggestion to incorporate structural and functional imaging in future experiments. We focused on the shorter timescale in the current study because we already saw a large shift after four days of MD (Figure 2h, k, m) and expected that the structural correlates would likely emerge within this time frame.

Minor:

4. In the abstract, the authors write “visual cortex activity following monocular deprivation remained stable”. This is not what they show, rather, there was an increase in open eye responses.

We have now changed the wording in the abstract to reflect our data more accurately.

5. Figure 1m: given these very low numbers, I am not sure whether one can claim that the number “was the same in both hemispheres”. At the very least, this should be phrased more carefully. How many cells were actually counted for this analysis?

We have now changed the wording in the text to reflect our data more accurately.

6. It should be indicated on how many sections, or um, the maximum intensity projections in Figure 3 are based on.

We have now added this information to the Figure 3 caption.

7. It is not clear what the right square bracket in Figure 4a indicates.

We have added text in the figure legend to clarify the meaning of the square brackets.

Referee #3 (Remarks to the Author):

This study’s aim is to determine if developmental myelination restricts neuronal plasticity. They do this by blocking oligodendrocyte maturation, thereby reducing myelination, in adolescent mice, and then subsequently assessing functional and structural experience-induced neuronal plasticity with monocular deprivation in the adult visual cortex. The authors demonstrate that blocking normal oligo developmental maturation results in adult visual cortical circuit’s response to visual stimulation being significantly weakened. This adult weakened circuit exhibits pronounced decreases in response following visual deprivation that are not found in control adult animals. This is the basis for the authors conclusion that there is a key role for adolescent oligodendrogenesis and myelin formation in restricting functional experience-dependent neuronal plasticity in the visual cortex.

Major Concern:

The authors have shown a clear effect of their developmental manipulation of oligodendrocytes on both the state of the cortical circuits and on their response to deprivation. But the conclusion that is drawn ‘myelination restricts plasticity’ implies that this is the only aspect of cortical organization that has been impacted by the manipulation, when it is clear that the functional synaptic organization of the circuit in animals that have undergone this manipulation is far from normal. This could be interesting if there was an effort to define how this manipulation was actually altering the circuit, but beyond ‘weaker responses’ and ‘lower spine density’, it is not clear exactly how the circuit has been altered, the mechanisms that underly this alteration, and how these alterations contribute to the effects of monocular deprivation that are seen. As it is presented, the only conclusion that can be derived is that altering oligodendrocyte maturation and myelination prevents the normal maturation of the cortex, and leaves the cortex in a state in which monocular deprivation exhibits an effect. This is far from strong, direct evidence that myelination restricts plasticity, which is the author’s claim.

We apologize for not being clearer in our messaging. We in fact, fully agree with the reviewer’s assessment that “altering oligodendrocyte maturation and myelination prevents the normal maturation of the cortex”, thus leading to enhanced functional and structural neuronal plasticity. We did not intend to imply that we have demonstrated a direct physical interaction between myelin and neurons that impairs their plasticity. Although this hypothesis is possible, we realize that we cannot experimentally test it with the limitations of current approaches. Rather, we believe – as the reviewer nicely summarized – that adolescent myelination is required for proper circuit maturation that leads to a decrease in plasticity, which does not occur when we genetically block oligodendrogenesis. However, it is also important to note, this is the first demonstration that altering oligodendrogenesis has such profound effects on functional experience-dependent neuronal plasticity, dendritic spine turnover, and inhibitory synaptic transmission. These results significantly increase our understanding of how developmental oligodendrogenesis and myelination impact cortical maturation and set the stage for more mechanistic studies in the future. We have modified the text throughout our manuscript to reflect this conclusion more accurately.

Other issues:

1. Previous studies have implicated inhibitory circuits as playing a significant role in regulating the period of visual cortex plasticity. While the authors show that the number of parvalbumin neurons, the candidate inhibitory cell type most expected to be involved, does not decrease, there is no direct test of the level of inhibition present in the altered circuit, and this remains one of many possible explanations for the presence of adult plasticity.

We thank the reviewer for bringing up this important point and agree that this is a likely circuit-level explanation for the increased plasticity we observed in *Myrf* cKO mice, given the large population of parvalbumin-expressing neurons that are myelinated in mouse cortex^{6,7}. We have now performed acute slice recordings in adult control and *Myrf* cKO mice and observed a drastic reduction in mIPSC

frequency, but not amplitude, in Myrf cKO mice. These results provide a circuit basis for the enhanced plasticity we observed in Myrf cKO mice and are now reported in Figure 5 and Extended Data Figure 11.

2. The measures of activity changes to visual stimulation were made with an indirect approach--intrinsic signal imaging—which measures changes in the oxy/deoxyhemoglobin ratio, i.e. a vascular change rather than a change in the activity of individual neurons. This seems odd given all the work of the last 15 years showing the greatly enhanced sensitivity and resolution of genetically encoded calcium sensors, which can be used at the episcala and at the cellular scale with 2-photon imaging to provide a much better measure of the neuronal activity patterns. Even stranger given that the authors used 2-P in vivo imaging to describe morphological changes.

The use of intrinsic signal imaging for determination of ocular dominance in mouse visual cortex has been validated repeatedly by comparison with the gold standard of action potential recordings from multiple neurons (from our laboratories in Cang et al, 2005⁸; Kaneko et al, 2008, 2014, 2017⁹⁻¹¹; and from many other laboratories, for example in PMIDs 37080344, 32711067, 31680800, 30257948, 28341863, 25662716, and 25583266). We used this well-established method because it is known to be sound for the purpose. In our hands, using genetically encoded calcium indicators, few neurons are scored as being well driven through both eyes in comparison to single cell extracellular recordings, and the calcium imaging distributions of ocular dominance are U-shaped rather than having a central mode that might be weighted toward one or the other eye, which is what is seen with extracellular action potential recordings. Given these caveats of calcium imaging for the specific application of assessing ocular dominance plasticity, we decided to use intrinsic signal imaging as a previously validated approach for our experiments.

3. The measurement of spine numbers here are from the apical dendrites of L5 pyramidal neurons. The authors need to address why L5 pyramidal neurons were chosen for observing plasticity changes in visual cortex.

Much of the existing literature describing spine dynamics in visual cortex examined layer V neurons¹²⁻¹⁹. Of note, the Thy1 reporter lines first described by Feng et al.²⁰ have commonly been used to study dendritic spine plasticity in visual cortex^{12-14,17-19,21}. Based on these previous studies, we believed that use of this line would allow us to readily resolve and track dendritic spines due to the relatively sparse and intense expression of the reporter, as well as capture a population of neurons that undergo functional plasticity following monocular deprivation. For this study, we used the Thy1-YFP-H line for in vivo imaging of dendritic spines in a subset of cortical pyramidal neurons. The line maintained in our laboratory has reporter expression in a mixture of deeper layer pyramidal neurons. We have added a low magnification image of YFP expression to Extended Data Figure 10.

4. The significance of the data displayed in the graphs of Figure 3d-h is not apparent. If the goal was to show differences in spine dynamics following deprivation between control animals and those with impaired oligodendrogenesis, the reader walks away with the exact opposite impression. Differences between control and knockout are present before and following monocular deprivation, so it is not clear why this analysis is of value in judging impact of monocular deprivation. Indeed, one change that appears to occur as a result of deprivation is an increase in the size of the spines in the control animals (which are not thought to exhibit plasticity) while the knockout animals (that are supposed to exhibit plasticity) appear to not increase in size.

We apologize for not making clear the significance of Figure 3. We did not have a specific goal, per se, to demonstrate a particular finding, though we did initially hypothesize that there may be a difference in the rate of spine turnover after sensory deprivation. Figure 3 describes what we observed at the level of structural synaptic plasticity, which is a significant increase in spine turnover even at baseline in the Myrf cKO mice. This enhanced structural plasticity implies a level of instability in circuit connectivity of Myrf cKO mice, both with and without any sensory manipulation. This is a novel finding and one that we believe is worth reporting, even though our hypothesis at the outset (that differences in spine turnover would emerge between phenotypes as a result of sensory deprivation) was incorrect.

4. Figure 2j- why is there an increase in the ipsi response in the control MD condition? This is not consistent with previous findings.

The increase we observed in ipsilateral responses following MD is consistent with previous findings in the literature that visual cortical plasticity in response to monocular deprivation in the adult consists primarily of an increase in response to the open eye rather than a prompt reduction in response to the deprived eye. See Sawtell et al., 2003¹⁷; Frenkel et al., 2004¹⁸; Sato & Stryker 2008, Kaneko et al., 2023¹⁹; and Craddock et al., 2023²⁰ for some examples. These references have also been added to the manuscript (Page 4, lines 158-159).

5. Figure 4 places changes in spine size in the context of dendritic topology, and it is really not clear why this analysis has been performed after the data in Figure 3 presents no clear picture of overall differences in spine morphology as a result of deprivation.

We apologize for not making clear the rationale behind the analysis in Figure 4. We observed, as reported in Figure 3h, a separation in spine size following four days of MD in control and cKO mice, which appeared to be driven by more spine size decreases in the cKO mice. Although it is true that the changes, when averaged, do not seem to deviate from baseline days, individual changes in synaptic weight, potentially resulting from the potentiation or weakening of a subset of inputs, can still influence the output of a circuit, especially when they are spatially clustered along the dendrite²⁶⁻²⁹. Given the practical implications of spatially clustered spine plasticity on circuit function, we performed the clustering analysis in Figure 4 to determine whether the changes in spine size were differentially

arranged along dendrites between control and cKO mice. We have now amended the text (Page 5, lines 190-195) to communicate the significance of this analysis more explicitly.

References

1. Young, K. M. *et al.* Oligodendrocyte Dynamics in the Healthy Adult CNS: Evidence for Myelin Remodeling. *Neuron* **77**, 873–885 (2013).
2. Wang, F. *et al.* Myelin degeneration and diminished myelin renewal contribute to age-related deficits in memory. *Nat. Neurosci.* 1–6 (2020) doi:10.1038/s41593-020-0588-8.
3. Hughes, E. G., Kang, S. H., Fukaya, M. & Bergles, D. E. Oligodendrocyte progenitors balance growth with self-repulsion to achieve homeostasis in the adult brain. *Nat. Neurosci.* **16**, 668–676 (2013).
4. Hughes, E. G., Orthmann-Murphy, J. L., Langseth, A. J. & Bergles, D. E. Myelin remodeling through experience-dependent oligodendrogenesis in the adult somatosensory cortex. *Nat. Neurosci.* **21**, 696–706 (2018).
5. Sato, M. & Stryker, M. P. Distinctive Features of Adult Ocular Dominance Plasticity. *J. Neurosci.* **28**, 10278–10286 (2008).
6. Micheva, K. D. *et al.* A large fraction of neocortical myelin ensheathes axons of local inhibitory neurons. *eLife* **5**, e15784 (2016).
7. Stedehouder, J. *et al.* Fast-spiking Parvalbumin Interneurons are Frequently Myelinated in the Cerebral Cortex of Mice and Humans. *Cereb. Cortex* **27**, 5001–5013 (2017).
8. CANG, J., KALATSKY, V. A., LÖWEL, S. & STRYKER, M. P. Optical imaging of the intrinsic signal as a measure of cortical plasticity in the mouse. *Vis. Neurosci.* **22**, 685–691 (2005).
9. Kaneko, M., Stellwagen, D., Malenka, R. C. & Stryker, M. P. Tumor necrosis factor- α mediates one component of competitive, experience-dependent plasticity in developing visual cortex. *Neuron* **58**, 673–680 (2008).
10. Kaneko, M. & Stryker, M. P. Sensory experience during locomotion promotes recovery of function in adult visual cortex. *eLife* **3**, e02798 (2014).
11. Kaneko, M., Fu, Y. & Stryker, M. P. Locomotion Induces Stimulus-Specific Response Enhancement in Adult Visual Cortex. *J. Neurosci. Off. J. Soc. Neurosci.* **37**, 3532–3543 (2017).
12. Zhou, Y., Lai, B. & Gan, W.-B. Monocular deprivation induces dendritic spine elimination in the developing mouse visual cortex. *Sci. Rep.* **7**, 4977 (2017).
13. Oray, S., Majewska, A. & Sur, M. Dendritic Spine Dynamics Are Regulated by Monocular Deprivation and Extracellular Matrix Degradation. *Neuron* **44**, 1021–1030 (2004).
14. Majewska, A. & Sur, M. Motility of dendritic spines in visual cortex in vivo: Changes during the critical period and effects of visual deprivation. *Proc. Natl. Acad. Sci.* **100**, 16024–16029 (2003).
15. Sajo, M., Ellis-Davies, G. & Morishita, H. Lynx1 Limits Dendritic Spine Turnover in the Adult Visual Cortex. *J. Neurosci.* **36**, 9472–9478 (2016).

16. Gökçe, O., Bonhoeffer, T. & Scheuss, V. Clusters of synaptic inputs on dendrites of layer 5 pyramidal cells in mouse visual cortex. *eLife* **5**, e09222 (2016).
17. Barnes, S. J. *et al.* Deprivation-Induced Homeostatic Spine Scaling In Vivo Is Localized to Dendritic Branches that Have Undergone Recent Spine Loss. *Neuron* **96**, 871-882.e5 (2017).
18. Kelly, E. A., Russo, A. S., Jackson, C. D., Lamantia, C. E. & Majewska, A. K. Proteolytic regulation of synaptic plasticity in the mouse primary visual cortex: analysis of matrix metalloproteinase 9 deficient mice. *Front. Cell. Neurosci.* **9**, (2015).
19. Majewska, A. K., Newton, J. R. & Sur, M. Remodeling of Synaptic Structure in Sensory Cortical Areas In Vivo. *J. Neurosci.* **26**, 3021–3029 (2006).
20. Feng, G. *et al.* Imaging Neuronal Subsets in Transgenic Mice Expressing Multiple Spectral Variants of GFP. *Neuron* **28**, 41–51 (2000).
21. Hofer, S. B., Mrsic-Flogel, T. D., Bonhoeffer, T. & Hübener, M. Experience leaves a lasting structural trace in cortical circuits. *Nature* **457**, 313–317 (2009).
22. Sawtell, N. B. *et al.* NMDA Receptor-Dependent Ocular Dominance Plasticity in Adult Visual Cortex. *Neuron* **38**, 977–985 (2003).
23. Frenkel, M. Y. & Bear, M. F. How Monocular Deprivation Shifts Ocular Dominance in Visual Cortex of Young Mice. *Neuron* **44**, 917–923 (2004).
24. Kaneko, M. & Stryker, M. P. Production of brain-derived neurotrophic factor gates plasticity in developing visual cortex. *Proc. Natl. Acad. Sci.* **120**, e2214833120 (2023).
25. Craddock, R., Vasalaukaite, A., Ranson, A. & Sengpiel, F. Experience dependent plasticity of higher visual cortical areas in the mouse. *Cereb. Cortex* **33**, 9303–9312 (2023).
26. Du, K. *et al.* Cell-type-specific inhibition of the dendritic plateau potential in striatal spiny projection neurons. *Proc. Natl. Acad. Sci.* **114**, E7612–E7621 (2017).
27. Colgan, L. A. *et al.* PKC α integrates spatiotemporally distinct Ca $^{2+}$ and autocrine BDNF signaling to facilitate synaptic plasticity. *Nat. Neurosci.* **21**, 1027–1037 (2018).
28. Harvey, C. D. & Svoboda, K. Locally dynamic synaptic learning rules in pyramidal neuron dendrites. *Nature* **450**, 1195–1200 (2007).
29. Losonczy, A. & Magee, J. C. Integrative Properties of Radial Oblique Dendrites in Hippocampal CA1 Pyramidal Neurons. *Neuron* **50**, 291–307 (2006).

Reviewer Reports on the First Revision:

Referees' comments:

Referee #1 (Remarks to the Author):

The revised manuscript from Xin and colleagues is substantially improved. The authors have carefully considered the points raised during the initial review, and they have thoughtfully revised the manuscript in response. Critical additional data has been incorporated into the manuscript that has strengthened the conclusions. The study provides important new insights into the role that developmental myelination plays in the maturation of neuronal circuits and in limiting neuronal plasticity. The work provides strong support for the important concept that developmental myelination significantly restricts neuronal plasticity. The manuscript provides critical new insights into CNS development.

Referee #2 (Remarks to the Author):

The authors have addressed all my comments, and the paper has been much improved.

Referee #3 (Remarks to the Author):

The authors have addressed a number of concerns raised in the previous review and added additional data showing that inhibition is one of the factors impacted by the manipulation of oligodendrogenesis and myelination in developing mouse cortex.

The authors responded to my concern that the previous version of the manuscript focused almost exclusively on the effects of the manipulation as 'enhancing' neuronal plasticity and have now acknowledged that the manipulation has a profound effect on the maturation of excitatory and inhibitory circuits.

Still, one wonders if what we are looking at here is essentially an immature circuit that, by virtue of this immaturity (in multiple factors including inhibition), is capable of the behaviors (such as plasticity) that immature circuits normally exhibit? In this regard it is notable that the baseline ocular dominance shown in Fig 2I for manipulated vs control animals is what one expects to see in early post eye-opening V1 neuronal responses that are strongly dominated by responses to the contralateral eye (Tan et al., 2022 J Neurosci 42, 3546-3556). This suggests that the increase in the number of neurons that respond to the ipsilateral eye and decrease in the numbers responsive to the contralateral eye that occurs after eye opening has not proceeded as a result of this manipulation. The immature eye dominance, the weak responsiveness, reduced inhibition, and the presence of plasticity are all consistent with the manipulation having left the circuit in an immature state. I recommend that the authors at least address this possibility 'impaired maturation' as an explanation for their findings in the Discussion.

Author Rebuttals to First Revision:

Referees' comments:

Referee #1 (Remarks to the Author):

The revised manuscript from Xin and colleagues is substantially improved. The authors have carefully considered the points raised during the initial review, and they have thoughtfully revised the manuscript in response. Critical additional data has been incorporated into the manuscript that has strengthened the conclusions. The study provides important new insights into the role that developmental myelination plays in the maturation of neuronal circuits and in limiting neuronal plasticity. The work provides strong support for the important concept that developmental myelination significantly restricts neuronal plasticity. The manuscript provides critical new insights into CNS development.

We are grateful to the reviewer for their excellent suggestions, which significantly improved our manuscript.

Referee #2 (Remarks to the Author):

The authors have addressed all my comments, and the paper has been much improved.

We are grateful to the reviewer for their excellent suggestions, which significantly improved our manuscript.

Referee #3 (Remarks to the Author):

The authors have addressed a number of concerns raised in the previous review and added additional data showing that inhibition is one of the factors impacted by the manipulation of oligodendrogenesis and myelination in developing mouse cortex.

We are grateful to the reviewer for their excellent suggestions, which significantly improved our manuscript.

The authors responded to my concern that the previous version of the manuscript focused almost exclusively on the effects of the manipulation as 'enhancing' neuronal plasticity and have now acknowledged that the manipulation has a profound effect on the maturation of excitatory and inhibitory circuits.

We are grateful to the reviewer for raising this point and believe the new experiments bring important additional information regarding the effect of preventing adolescent oligodendrogenesis on inhibitory circuit maturation.

Still, one wonders if what we are looking at here is essentially an immature circuit that, by virtue of this immaturity (in multiple factors including inhibition), is capable of the behaviors (such as plasticity) that immature circuits normally exhibit? In this regard it is notable that the baseline ocular dominance shown in Fig 2I for manipulated vs control animals is what one expects to see in early post eye-opening V1 neuronal responses that are strongly dominated by responses to the contralateral eye (Tan et al., 2022 J Neurosci 42, 3546-3556). This suggests that the increase in the number of neurons that respond to the ipsilateral eye and decrease in the numbers responsive to the contralateral eye that occurs after eye opening has not proceeded as a result of this manipulation. The immature eye dominance, the weak responsiveness, reduced inhibition, and the presence of plasticity are all consistent with the manipulation having left the circuit in an immature state. I recommend that the authors at least address this possibility 'impaired maturation' as an explanation for their findings in the Discussion.

We thank the reviewer for bringing up this perspective and have added additional text in the discussion addressing this point (lines 242-244).